# Structural insights into FSP1 catalysis and ferroptosis inhibition

Yun Lv[1,7], Chunhui Liang[2,7], Qichao Sun[1,7], Jing Zhu[3,7], Haiyan Xu[1], Xiaoqing Li[1], Yao-yao Li[4], Qihai Wang[5] ✉, Huiqing Yuan [1,6] ✉, Bo Chu [2] ✉ & Deyu Zhu [1] ✉

Ferroptosis suppressor protein 1 (FSP1, also known as AIMF2, AMID or PRG3) is a recently identified glutathione-independent ferroptosis suppressor[1-3], but its underlying structural mechanism remains unknown. Here we report the crystal structures of *Gallus gallus* FSP1 in its substrate-free and ubiquinone-bound forms. The structures reveal a FAD-binding domain and a NAD(P)H-binding domain, both of which are shared with AIF and NADH oxidoreductases[4-9], and a characteristic carboxy-terminal domain as well. We demonstrate that the carboxy-terminal domain is crucial for the catalytic activity and ferroptosis inhibition of FSP1 by mediating the functional dimerization of FSP1, and the formation of two active sites located on two sides of FAD, which are responsible for ubiquinone reduction and a unique FAD hydroxylation respectively. We also identify that FSP1 can catalyze the production of $H_2O_2$ and the conversion of FAD to 6-hydroxy-FAD in the presence of oxygen and NAD(P)H in vitro, and 6-hydroxy-FAD directly inhibits ferroptosis in cells. Together, these findings further our understanding on the catalytic and ferroptosis suppression mechanisms of FSP1 and establish 6-hydroxy-FAD as an active cofactor in FSP1 and a potent radical-trapping antioxidant in ferroptosis inhibition.

Ferroptosis is an iron-dependent form of regulated cell death due to excessive lipid peroxidation[10,11], and implicated in a variety of pathological conditions and diseases, and the treatment of cancer[11-13]. Ferroptosis has attracted tremendous interest because it has become a highly promising target for treating cancer. Until now, ferroptosis is regulated by at least four major redox-related systems, which include glutathione-GPX4[14], FSP1-coenzyme Q10 ($CoQ_{10}$)[2,3], DHODH-$CoQ_{10}$[15] and GCH1-tetrahydrobiopterin (BH4) pathways[16]. Among of them, FSP1-$CoQ_{10}$ is an independent parallel system that cooperates with the primary glutathione-GPX4 pathway to inhibit ferroptosis. FSP1 is recently identified as a breakthrough owing to its unique ability to suppress ferroptosis by reducing ubiquinone to ubiquinol[2,3], or

vitamin K to reduced hydroquinone (VKH2)[17]. FSP1 is previously known as apoptosis-inducing factor mitochondrial 2 (AIFM2, also known as AMID or PRG3) on the basis of its homology with apoptosis-inducing factor (AIF or AIFM1)[18,19], and acts as an NAD(P)H-dependent oxidoreductase in vitro[20]. Unlike AIF, FSP1 lacks substantial pro-apoptotic function, but protects cells from ferroptosis. More recently, FSP1 was demonstrated to regulate NADH redox during thermogenesis in brown adipocytes[21]. Although accumulating evidence have clearly demonstrated that FSP1 acts as a ferroptosis suppressor, it is still not fully understood how FSP1 catalyzes the reduction of ubiquinone by NAD(P)H and defends against ferroptosis through its oxidoreductase activity, and why only FSP1 in AIF family plays a key role in suppressing

[1]Department of Biochemistry and Molecular Biology, School of Basic Medical Sciences, Cheeloo College of Medicine, Shandong University, Jinan 250012, China. [2]Department of Cell Biology, School of Basic Medical Sciences, Cheeloo College of Medicine, Shandong University, Jinan 250012, China. [3]State Key Laboratory of Microbial Technology, Shandong University, Qingdao 266237, China. [4]Key Laboratory of Chemical Biology (Ministry of Education), School of Pharmaceutical Sciences, Cheeloo College of Medicine, Shandong University, Jinan 250012, China. [5]School of bioengineering, Jingchu University of Technology, Jingmen 448000, China. [6]Key Laboratory of Experimental Teratology of Ministry of Education, Institute of Medical Sciences, the Second Hospital, Cheeloo College of Medicine, Shandong University, Jinan 250031, China. [7]These authors contributed equally: Yun Lv, Chunhui Liang, Qichao Sun, Jing Zhu. ✉e-mail: wangqihai@jcut.edu.cn; lyuanhq@sdu.edu.cn; chubo123@sdu.edu.cn; zhudeyu@sdu.edu.cn

ferroptosis. Exploring the molecular mechanisms of FSP1 catalytic activity will be beneficial for developing therapeutics that acts through this FSP1-CoQ$_{10}$ system.

The human and chicken (*Gallus gallus*) FSP1 (hereafter named hFSP1 and cFSP1 respectively), which we have studied here, share 70% sequence identities (Supplementary Fig. 1), and are characterized by a short N-terminal myristoylation motif that is required for lipid droplets formation and membrane targeting[2,3], and a canonical flavin adenine dinucleotide (FAD)-dependent oxidoreductase domain that is responsible for the catalytic activity[22]. In the present study, we initially attempted to obtain the crystal of hFSP1, but were not successful. Therefore, cFSP1 was purified and crystallized. We report the crystal structures of cFSP1 (residues 12–373, hereafter named cFSP1$^{\Delta N}$) in its substrate-free and ubiquinone (CoQ$_1$)-bound forms at 2.0 and 2.6 Å resolution (Supplementary Table 1), respectively. The representative images of the electron density are shown in Supplementary Fig. 2. Our work reveals that the homodimer of FSP1 is required for the formation of FAD- and ubiquinone-binding pockets, as well as the activity of NAD(P)H: ubiquinone oxidoreduction and ferroptosis suppression. The formation of FSP1 homodimer is dependent on its C-terminal domain (CTD), deficiency of which is unable to form the functional homodimer of FSP1 and loses the protective effect to inhibit ferroptosis. Moreover, we also identify 6-hydroxy-FAD as an active cofactor of FSP1 and a potent anti-ferroptotic compound. Taken together, our study elucidates the role of FSP1 catalytic activity in suppressing ferroptosis and provides new insight into drug design and screening.

## Results and discussion

### The cFSP1 structure adopts a glutathione reductase-like fold with a unique carboxy-terminal domain

For crystallization, the short N-terminal hydrophobic sequence (residues 1–11, myristoylation motif) of cFSP1 was cut off and the protein was methylated using a reductive lysine methylation protocol[23]. The cFSP1$^{\Delta N}$ lacks only the myristoylation motif, and almost resembles full-length cFSP1 in structure. The overall structure of cFSP1$^{\Delta N}$ adopts a glutathione reductase-like fold and is also composed of three domains, a FAD-binding domain (FBD, residues 12–141, and 241–318) harboring a FAD molecule, a NAD(P)H-binding domain (NBD, residues 142–240) and a C-terminal domain (CTD, residues 319–373) (Fig. 1a–c and Supplementary Fig. 1). Both FBD and NBD display the classical Rossmann fold and form the core of the FAD-dependent oxidoreductase domain, whereas the CTD contains two β-strands (β17- β18) and one α-helix (α11), and locates on the side of the FAD-dependent oxidoreductase domain. Structural alignment using Dali server[24] revealed that cFSP1 is structurally homologous to bacterial NDH-2 enzymes, yeast NDH-2 (Ndi1)[7,25], human AIF[5] and other NADH dehydrogenases, and the closest structural homolog is an NDH-2 from *Caldalkalibacillus thermarum*[4,8,9], although they all have a sequence identity of less than 26%. The superposition of the structures of cFSP1 and NDH-2 from *Caldalkalibacillus thermarum* highlights their overall similarity in the FBD and NBD structures (Fig. 1d, e), with a root mean squared deviation (r.m.s.d.) of 2.4 Å when compared with FBD and NBD, consistent with both having NADH oxidase activity. However, there is an obvious difference in the CTD structure, which is highly conserved in FSP1 proteins from different species (Supplementary Fig. 1), indicating that the CTD may be related to the distinct functions of FSP1 and other FAD-dependent NADH dehydrogenases.

### The cFSP1 structures form a functional homodimer

The structure of the ubiquinone-bound form was obtained by co-crystallizing cFSP1$^{\Delta N}$ with ubiquinone Q1 (CoQ$_1$) in the absence of NAD(P)H (Fig. 1c). In both substrate-free and CoQ$_1$-bound structures, each asymmetric unit contains one cFSP1$^{\Delta N}$ molecule, and their structures are virtually identical (r.m.s. deviation of ~0.4 Å), with the exception of a loop 327–337 at CTD, which assumes two different

conformations even though it is partially disordered and displays weak density (Supplementary Fig. 3), suggesting a role in regulating substrate access to the active site. Notably, in these two structures, cFSP1$^{\Delta N}$ molecule forms a dimer with its crystallographic symmetric molecule (Fig. 2a, b). The interface between the cFSP1$^{\Delta N}$ monomers is extensive, with a buried surface area of 7574.0 Å$^2$, which accounts for ~ 25% of the total solvent-accessible surface area of the cFSP1$^{\Delta N}$ dimer as calculated by the PDBePISA serve[26]. The interactions between the two cFSP1 monomers are mainly mediated through its CTD. The two β-strands (β17- β18) of CTD from each monomer protrudes into a hole found in the surface of the other monomer to form part of the active pocket, and a four-stranded β-sheet structure with characteristics of a β-barrel-like arrangement (Fig. 2c). Accordingly, the results of gel filtration chromatography indicated that cFSP1, cFSP1$^{\Delta N}$ and methylated cFSP1$^{\Delta N}$ mainly exist as dimers in solution (Supplementary Fig. 4a, b), consistent with the dynamic light scattering (DLS) results (Supplementary Fig. 4c), suggesting that cFSP1 functions as a dimer.

In addition, unlike AIF[27,28], the dimerizations of both cFSP1 and hFSP1 are not mediated by NADH (Supplementary Fig. 4c). Interestingly, the oligomeric state of hFSP1 is dynamic and diverse in vitro, the gel filtration chromatography showed that hFSP1 behaved as mostly monomer (Supplementary Fig. 4a), DLS results showed that hFSP1 sometimes started as a monomer, then automatically and fast dimerized at low temperature, while hFSP1 normally existed as a dimer at either low or room temperature (Supplementary Fig. 4c). Although the slight difference between gel filtration and DLS results allows a little possibility that hFSP1 may be monomer during catalysis, our data suggested that hFSP1 functions as a dimer similar to cFSP1, which is further supported by co-IP results showing that hFSP1 formed a dimer with itself in cells (Fig. 2h). Indeed, homo-oligomerization is a common feature among membrane-associated protein structures, and similar homodimer structures were observed in crystals of AIF, Ndi1 and bacterial NDH-2 enzymes[4–9,25]. Therefore, the homodimer observed in cFSP1 structures should be physiologically relevant and the dimerization should be required for FSP1 function.

### The CTD plays a key role in the function of FSP1

Typically, a myristoyl anchor is insufficient for membrane targeting, and other factors are required[29]. We found that the dimerization of FSP1 can enhance membrane binding by a dual myristoyl anchor. That is, the crystallographic two-fold axis relating the monomers within the dimer is perpendicular to the membrane, the N-terminal myristoylation motif serves as two-pronged membrane anchors, and the CTD may also be involved in membrane association, consistent with a model calculated using PPM server[30] (Supplementary Fig. 5). This is also presumed that the side containing two quinone tunnels, which are mainly formed by α10-β17 loop, β18-α11 loop and helix α11 of the CTD, adjacent to the membrane surface for facilitating the entry of lipophilic substrate from the membrane (Fig. 2b and Supplementary Fig. 5). In contrast, another opposite side is predominantly charged and predisposed to be the cytoplasmic face of the FSP1 dimer (Fig. 2a and Supplementary Fig. 5).

It has been reported that hFSP1 mutant deleted C-terminal 73 amino acid residues (residues 1–300) was exclusively detected in the nucleus fraction[19], supporting a role for the CTD in FSP1 membrane anchoring. We further confirmed the functional importance of the CTD through enzyme assay and rescue assay. The deletion constructs of CTD (cFSP1$^{\Delta CTD}$, residues 1–318, or hFSP1$^{\Delta CTD}$, residues 1–316) abolished the NADH oxidation in vitro and hFSP1$^{\Delta CTD}$ obviously decreased the ferroptosis inhibition activities in cells (Fig. 2d–g). Interestingly, although both cFSP1$^{\Delta CTD}$ and hFSP1$^{\Delta CTD}$ mainly exist as non-functional dimers in solution (Supplementary Fig. 4), it is supposed that the non-functional dimerization mode of FSP1$^{\Delta CTD}$ is different from that of FSP1 full-length, which is a functional dimerization mode and dependent on the CTD. In short, CTD is involved in FSP1 functional

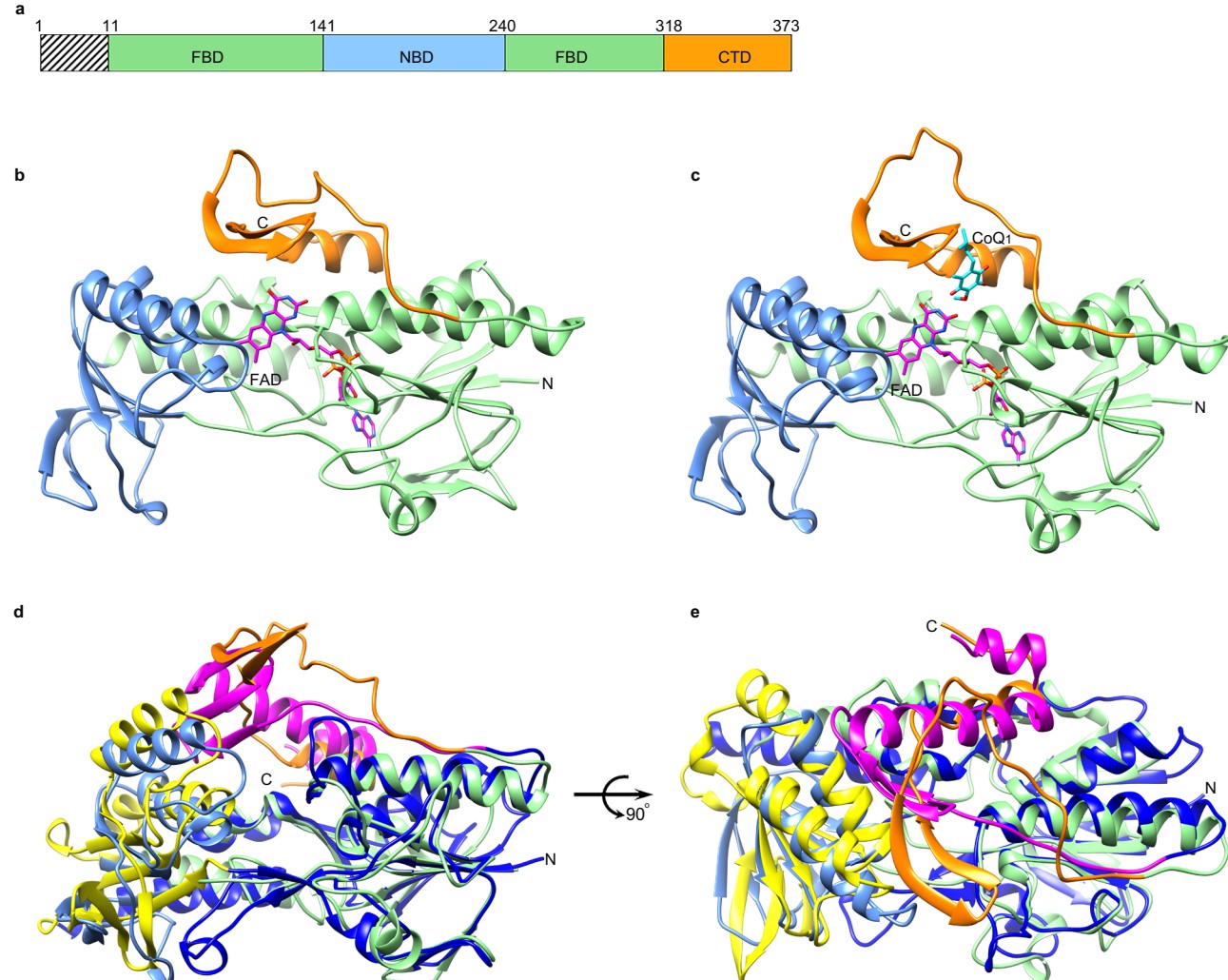

**Fig. 1 | Overall structures of cFSP1. a** Schematic representations of cFSP1. N-terminal myristoylation motif excluded in crystallized cFSP1 is marked with hatched lines. Lime, blue and orange boxes denote the FBD (residues 12–141 and 241–318), NBD (residues 142–240) and CTD (residues 319–373), respectively. **b, c** Cartoon representation of cFSP1$^{\Delta N}$ in substrate-free form (**b**) and in complex with CoQ$_1$ (**c**). Individual cFSP1$^{\Delta N}$ domains are colored according to the scheme in (**a**). FAD and CoQ$_1$ molecules are shown as stick representations. **d, e** Two views of the structural alignment between cFSP1 (substrate-free form) and NDH-2 from *Caldalkalibacillus thermarum* (PDB, 4NWZ). The FBD, NBD and CTD of NDH-2 are coloured in marine, yellow and pink, respectively. Domains of cFSP1 are also colored according to the scheme in (**a**).

homodimerization, membrane association, and oxidoreductase activity, and thus plays important multiple roles in the function of FSP1.

## The CTD mediates the formation of FAD- and ubiquinone-binding pockets

In our structures, the FAD cofactor is clearly defined, binding in an elongated manner, which is similar to that observed in the structures of AIF, Ndi1, and bacterial NDH-2s. The adenine and pyrophosphate groups of the FAD interact with the conserved region of the FBD (residues 17–21, 40–42, 48, 81, 107–109, 119, 247–250, and 283–284), and the isoalloxazine ring of the FAD is positioned at the intersection of FBD, NBD and CTD domains (Fig. 3a, b).

Notably, the isoalloxazine ring of the FAD in cFSP1 rotates ~ 25 degrees compared with that in all known structures of other FAD-dependent oxidoreductases, while the other isoalloxazine rings are basically in the same plane (Supplementary Fig. 6a), indicating a unique feature of FSP1 structure. The side chains of Val49 and Tyr343 stack to two sides of the middle ring of the isoalloxazine ring, making hydrophobic interactions, respectively. The peteridine moiety of the isoalloxazine ring is located in a canonical active site for ubiquinone substrate binding (hereafter named ubiquinone-binding

pocket), which is mainly formed by residues 293–298 of α10, residues 327–331 and 354–362 from CTD, residues 340–350 from CTD of the other chain in the dimer, where one CoQ$_1$ molecule is clearly buried in (Figs. 1c and 3c). The CoQ$_1$ interacts with cFSP1 by mainly hydrophobic interactions with residues forming the ubiquinone-binding pocket, including Ala294, Tyr295, residues 327–329, Phe359, Lys362 and two residues (Val340 and Leu348) from the other chain of the dimer (Fig. 3c). The truncation of C-terminal 359–373 amino acid residues (cFSP1$^{\Delta C15}$, residues 1–358) almost abolished the NADH oxidation (Supplementary Table 2), and the truncation of C-termianl 360–373 amino acid residues in hFSP1 (hFSP1$^{\Delta C14}$, residues 1–359) decreased the ferroptosis inhibition activities (Fig. 2e, g). Moreover, the head group of CoQ$_1$ interacts with the FAD through a hydrogen bond between the O4 atom of CoQ$_1$ and N3 atoms of FAD (Fig. 3b, c).

Unexpectedly, the xylene moiety of the isoalloxazine ring is also located in another smaller solvent-accessible pocket (hereafter named FAD hydroxylation pocket), which is formed by residues 151–155 at the N-terminus of α 6, residues 49–53, residues Lys354 and Leu358 from CTD, residues 343–345 from CTD of the other chain in the dimer (Fig. 3d and Supplementary Fig. 6b). This FAD hydroxylation pocket is only found in FSP1 structure, but not in the

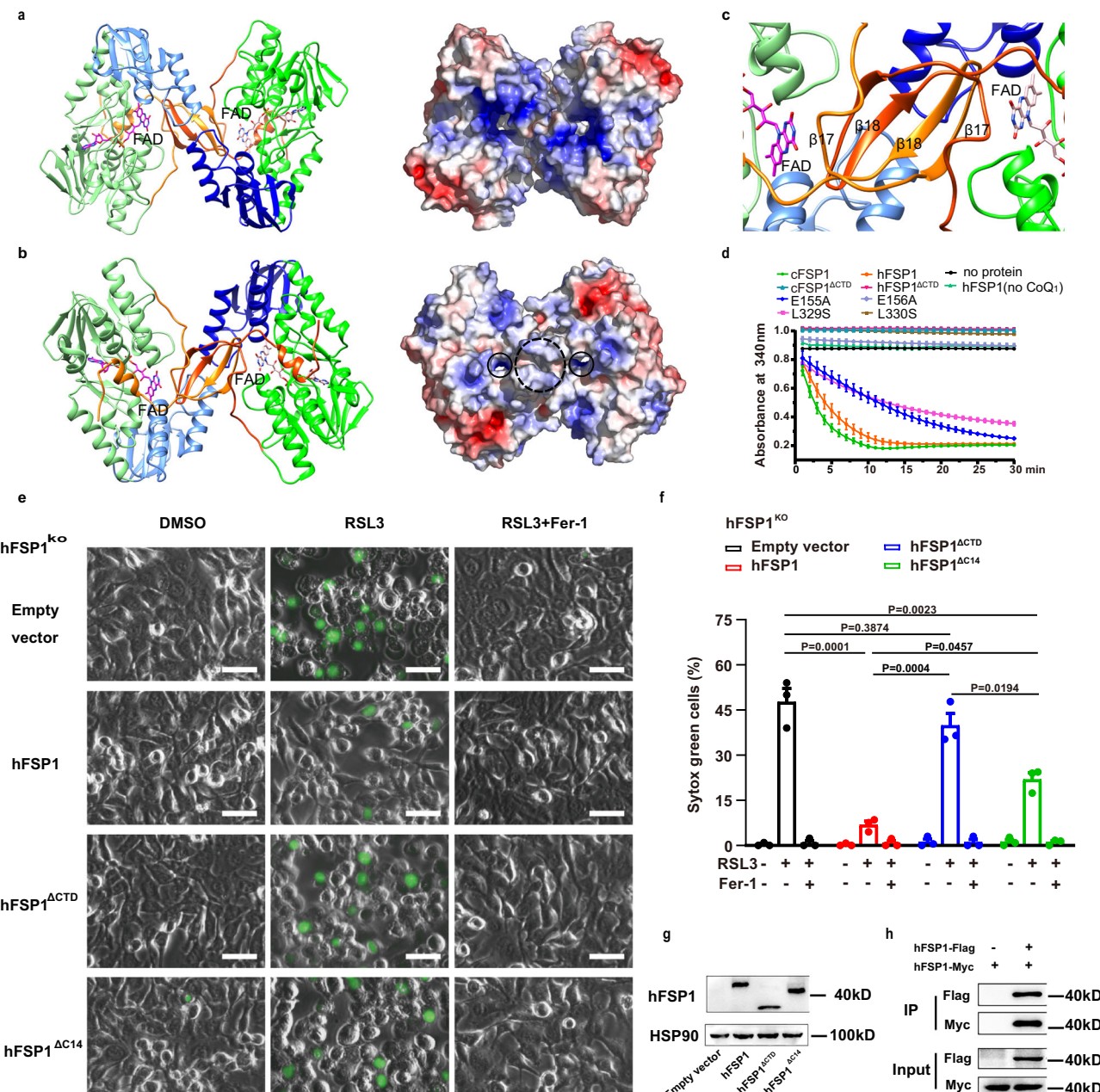

**Fig. 2 | The homodimer and CTD of cFSP1^ΔN. a, b** The cFSP1^ΔN homodimer is viewed along its crystallographic two-fold axis and from above (**a**) or below (**b**) the membrane plane. Left panels show cartoon representations of the dimeric structure, one portomer is colored as in Fig. 1b and the other one is coloured in deep green (FBD), marine (NBD), and orange red (CTD). The FAD cofactors are shown as stick representations. The right panels show the ± 5 kT/e electrostatic potential surfaces of the dimer, with surfaces of negative potential in red, positive in blue, and uncharged in white. The approximate regions for membrane association and the quinone tunnels are highlighted with black dashed and solid circles, respectively. **c** A close-up view of two CTDs of the dimer, which form a β-barrel-like arrangement. **d** NADH consumption assay (340 nm) using recombinant cFSP1 (0.5 μM) or hFSP1 (0.2 μM) as indicated. cFSP1^ΔCTD, residues 1–318 in cFSP1; hFSP1^ΔCTD and hFSP1^ΔC14 represent residues 1–316 and 1–359 in hFSP1, respectively. **e** Representative images of HT1080 hFSP1^KO cells transfected with the indicated

hFSP1 constructs, and then treated with DMSO, RSL3 (75 nM) or with RSL3 (75 nM) and Fer-1 (2 μM) for 3 h. Dead cells were stained by Sytox Green. Scale bars, 50 μm. Representative results from one of three experiments are shown. **f** Cell death analysis of the cell lines depicted in (**e**), statistical significance of differences between two different groups were analyzed by two-sided one-way ANOVA with Tukey's test. $P$ values indicated (95% CI of diff., Empty vector $vs$ hFSP1, hFSP1^ΔCTD, hFSP1^ΔC14: 26.03% to 55.58%, −6.98% to 22.58%, 10.96% to 40.51%; hFSP1 $vs$ hFSP1^ΔCTD, hFSP1^ΔC14: −47.78% to −18.23%, −29.85% to −0.30%; hFSP1^ΔCTD $vs$ hFSP1^ΔC14: 3.16% to 32.71%). **g** Western blot analysis of HT1080 hFSP1^KO cells expressing the indicated proteins, which were independently repeated twice with similar results. **h** Western blot analysis of Co-IP experiment using HEK293T cells co-transfected with hFSP1-Flag and hFSP1-Myc, which was perfomed only once. Data in **d** and **f** represent the mean ± s.e.m. of three experiments ($n$ = 3). Source data are provided as a Source Data file.

structures of other known NADH:quinone oxidoreductases, such as AIF, NDH-2 and Ndi1 (Supplementary Fig. 6b). Mutations of the residues relating to FAD-binding, ubiquinone-binding or FAD hydroxylation pockets reduced the NADH oxidation activity of FSP1 proteins (Supplementary Table 2), among which K117A, K292A/ M293A/Y295A, R53A mutations of cFSP1 were almost completely inactive in vitro.

Although we failed to obtain the crystal structure of cFSP1 in complex with NAD(P)H, the high conservation of the NADH binding mode in NADH:quinone oxidoreductases readily allowed the bound

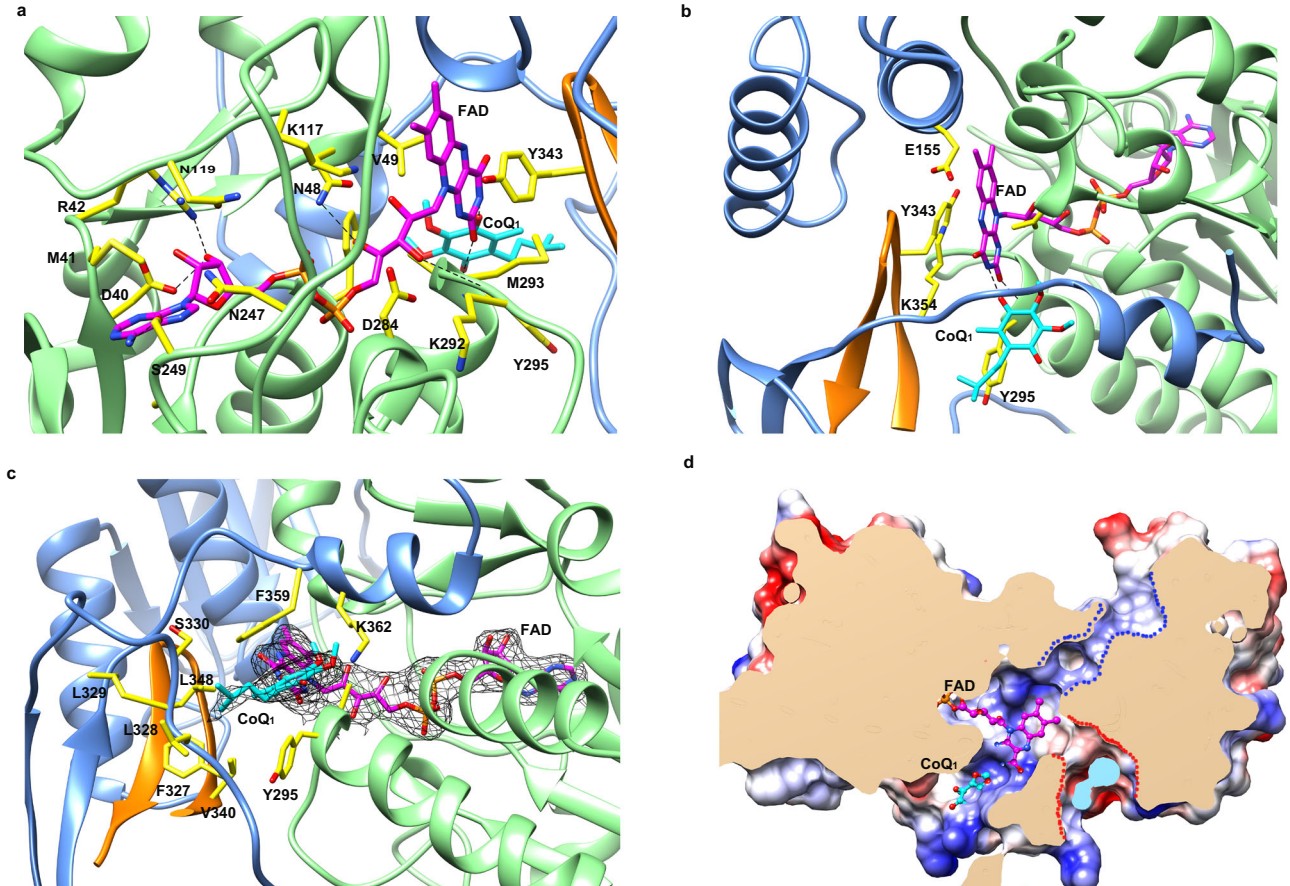

**Fig. 3 | FAD- and ubiquinone-binding pockets in the structures of cFSP1$^{\Delta N}$ in complex with CoQ$_1$. a** FAD-binding pocket, the FAD and FAD-contacting residues are labeled and shown as stick representations, and the nitrogen and oxygen atoms are coloured in blue and red, respectively. The carbon atoms of the FAD and FAD-contacting residues are coloured in pink and yellow, respectively, the hydrogen bonds are shown in black dashed lines. **b** Interactions between the isoalloxazine ring of the FAD and the cFSP1 protein, residues Val49, Glu155, Lys354, Tyr295, and Tyr343 from the other monomer of the dimer are labeled and shown as stick representations. **c** Ubiquinone-binding pocket, the CoQ$_1$ and the residues lining the

pocket are labeled and shown as stick representations. The 2$Fo - Fc$ electron density (black) is contoured at 1.0 σ and shown for FAD and CoQ$_1$. **d** Clipped surface representation of the FSP1 dimer showing the pockets in one monomer. The clipped interior surfaces of the two monomers were coloured as wheat and light blue, respectively. The FAD-binding and CoQ-binding pockets were occupied by FAD (pink carbon atoms) and CoQ$_1$ (cyan carbon atoms), respectively. The NAD(P)(H)-binding and FAD hydroxylation pockets were indicated by blue and red dashed lines, respectively.

NADH observed in *Caldalkalibacillus thermarum* NDH-2 to be modeled into cFSP1 in NAD(P)H-binding pocket of NBD, with the nicotinamide ring approaching the re-face of the FAD and a conformation change of Tyr343 of the other chain in cFSP1 dimer in order to avoid a steric clash (Supplementary Fig. 7a, b). We propose that Tyr343 may also be involved in the NADH-induced homodimerization and cFSP1 activity regulation, supporting by that Y343A and Y343F mutations reduced its NADH oxidation activity (Supplementary Table 2). Based on the model of hypothetical cFSP1-NAD+complex (Supplementary Fig. 7b, c), cFSP1 has a different interacting region for AMP portion of NAD + . Unlike NDH-2 and Ndi1 contain Glu at this position (Glu198 of NDH-2, Glu273 of Ndi1), which interacts with adenine ribose moiety of NAD and thus determines Ndi1 or NDH-2 selectivity for NADH over NADPH[8,25], cFSP1 contain His173 (His174 for hFSP1) at the corresponding position and would have a different selectivity comparing with Ndi1 and NDH-2. Indeed, our data demonstrate that hFSP1 oxidized NADH and NADPH with similar efficiency and kinetics ($K_m$, $k_{cat}$) in combination with CoQ$_1$ (Supplementary Fig. 7d and Table 3), and $K_m$ Value for CoQ$_1$ reduction due to NADH oxidation was similar to that due to NADPH oxidation, indicating that the purified FSP1 proteins have no obvious preference for NADH or NADPH oxidation.

## FSP1 has a complex reaction involved in ferroptosis inhibition

We used two mutations to examine the role of FSP1 CoQ oxidoreductase activity in suppressing ferroptosis. The E155A mutation in cFSP1 (E156A in hFSP1) was reported to greatly impair the CoQ oxidoreductase activity and abolish the ability of ferroptosis inhibition[2]. The mutation of Leu329 (Leu330 in hFSP1), which is positioned at the entrance loop for the substrate CoQ$_1$ (Fig. 3c), probably reduces the binding activity to CoQ. As expectedly, both the E155A cFSP1 and E156A hFSP1 mutations obviously reduced the CoQ oxidoreductase activity, especially the E156A hFSP1 mutation is completely inactive in the NADH oxidation assay using CoQ$_1$ as substrate and cell death assay (Fig. 2d, Fig. 4a and Supplementary Table 2). The second mutation L329S (L330S in hFSP1) also obviously impaired its CoQ oxidoreductase activity (Fig. 2d and Supplementary Table 2). The effect of mutation L329S (L330S in hFSP1) on CoQ oxidoreductase activity was almost same as that of mutation E155A (E156A in hFSP1), the L330S hFSP1 mutation completely lost the CoQ oxidoreductase activity as E156A hFSP1 mutation. Interestingly, the L330S hFSP1 mutation can still partially rescue the resistance of HT1080 hFSP1$^{KO}$ cells to RSL3, and significantly higher than that of mutation E156A (Fig. 4), indicating that the ferroptosis inhibition of FSP1 may not only depend on its CoQ oxidoreductase activity. In addition, our structures revealed that there

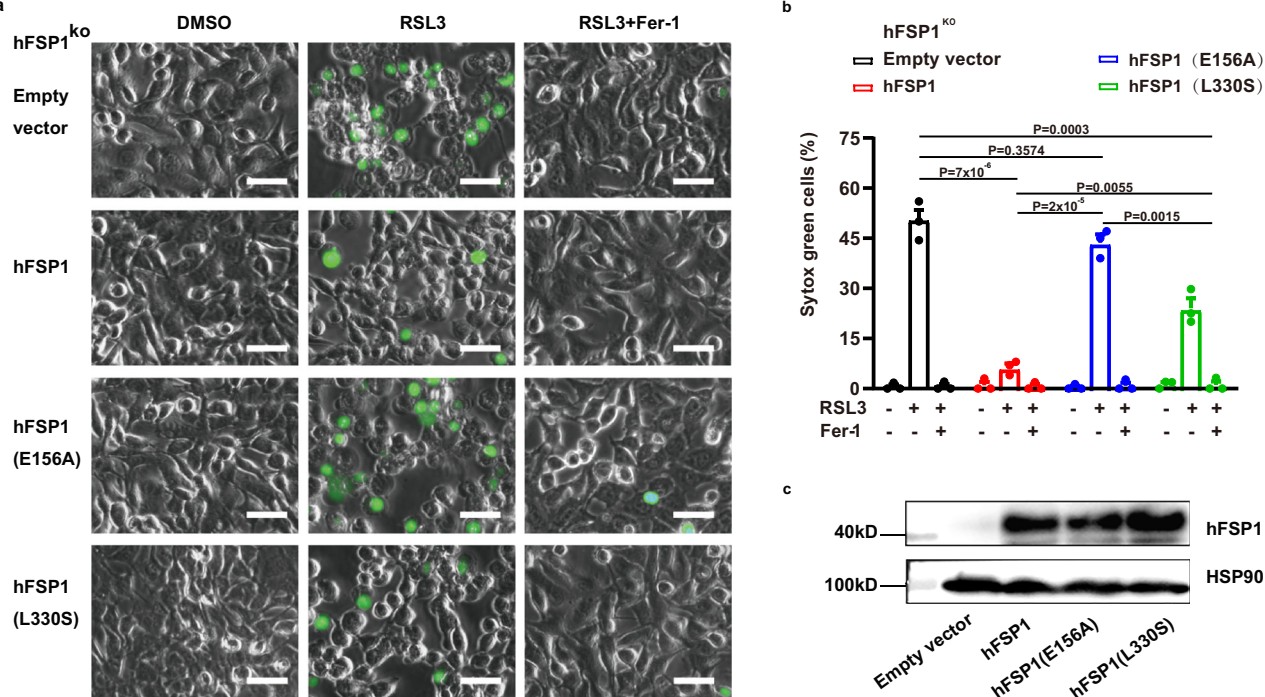

**Fig. 4 | Anti-ferroptotic analysis of hFSP1 mutants. a** Representative images of HT1080 hFSP1[KO] cells transfected with the indicated hFSP1 constructs, and then treated with the indicated reagents for 3 h. Dead cells were stained by Sytox Green. Scale bars, 50 μm. Representative results from one of three experiments are shown. **b** Cell death analysis of the cell lines depicted in (**a**), statistical significance of differences between two different groups were analysed by two-sided one-way ANOVA with Tukey's test. *P* values indicated (95% CI of diff., Empty vector *vs* hFSP1,

E156A, L330S: 32.38% to 54.48%, −4.99% to 17.12%, 15.62% to 37.72%; hFSP1 *vs* E156A, L330S: −48.42% to −26.32%, −27.82% to −5.72%; E156A *vs* L330S: 9.55% to 31.65%). Data represent the mean ± s.e.m. of three experiments (*n* = 3). **c** Western blot analysis of HT1080 hFSP1[KO] cells expressed the indicated hFSP1 proteins, the western blot was independently repeated twice with similar results. Source data are provided as a Source Data file.

is no strong interaction between Glu155 with FAD, and Glu155 is not located in the CoQ-binding pocket (Fig. 3b), suggesting that E155A mutation (E156A in hFSP1) impairs its function not only by decreasing the affinity for FAD and CoQ but also by affecting other factors, such as the previous studies that *Staphylococcus aureus* NDH-2 Glu172 at this position plays a key role in proton transfer during the catalytic reaction[31,32]. Moreover, our structures support that FSP1 would has similar enzymatic characteristics to NDH-2, whose catalytic cycle has been described as composed of two half reactions, including NADH oxidation/flavin reduction and flavin re-oxidation / quinone reduction[8]. Therefore, we propose that FSP1 has similar complex reaction to be involved in ferroptosis inhibition, and further explore the catalytic activity of FSP1.

### The recombinant FSP1 proteins contain both FAD and 6-hydroxy-FAD cofactors

The hFSP1 contains modified flavin 6-hydroxy-FAD cofactor, which has a green color and is thought to be a result of a specific modification of the FSP1-bound flavin during protein expression in vivo[20]. Indeed, we observed that the obtained cFSP1, hFSP1, native and methylated cFSP1[ΔN] from *E. coli* had a green color before and after purification, indicating the presence of 6-hydroxy-FAD, which was also confirmed by HPLC-MS/MS (Fig. 5a, b, and Supplementary Fig. 8) and NMR analyses (Supplementary Fig. 9). Furthermore, HPLC analysis also showed that the flavin cofactor from the recombinant FSP1 proteins was a mixture of 6-hydroxy-FAD and FAD, and that the dominant flavin cofactor of the recombinant cFSP1 is FAD, while the dominant flavin cofactor of the recombinant hFSP1 is 6-hydroxy-FAD (Fig. 5b). In addition, we speculate that the exact nature at the 6 positions of the modified flavin is hydroxyl, although another isomer including ketonyl at the 6 positions cannot be completely ruled out (Supplementary

Figs. 9 and 10). To explore the effect of FAD or 6-hydroxy-FAD on the NAD(P)H oxidation activity, apo-hFSP1 was prepared by removing all flavin cofactors from the recombinant hFSP1, and the NADH oxidation activity was determined after reconstitution with FAD or 6-hydroxy-FAD. The NADH oxidation assay showed that the initial reaction velocity of 6-hydroxy-FAD-reconstituted hFSP1 was fast similar to that of native purified hFSP1, but decreased rapidly after 3 min, which maybe resulted from the poor stability of 6-hydroxy-FAD-reconstituted hFSP1, while the initial reaction velocity of FAD-reconstituted hFSP1 was obviously slower than that of 6-hydroxy-FAD-reconstituted hFSP1, but almost constant until the reaction was complete (Supplementary Fig. 11a). Furthermore, our enzyme kinetic assays showed that although 6-hydroxy-FAD-reconstituted hFSP1 exhibited lower affinities ($K_m$) for NADH and CoQ$_1$ than FAD-reconstituted hFSP1, the NADH oxidation activity ($k_{cat}$) of 6-hydroxy-FAD-reconstituted hFSP1 was higher than that of FAD-reconstituted hFSP1 (Supplementary Table 3). These data suggest that the 6-hydroxy-FAD-containing FSP1 is a catalytically active enzyme, and its NAD(P)H oxidation activity is higher than that of FAD-containing FSP1, which may also explain that the initial reaction rate of hFSP1 with more 6-hydroxy-FAD was clearly faster than that of cFSP1 with less 6-hydroxy-FAD (Supplementary Fig. 11b).

### FSP1 proteins generate 6-hydroxy-FAD by FAD hydroxylation pocket

The hFSP1-bound FAD can be converted to 6-hydroxy-FAD by hFSP1 during aerobic turnover with NADPH in vitro[20]. We confirmed that both FAD-reconstituted hFSP1 and cFSP1 could catalyze the conversion of FSP1-bound FAD to 6-hydroxy-FAD in the presence of oxygen and NAD(P)H in vitro, and the reaction was promoted by hydrogen peroxide (Fig. 5c). According to our structures, it is easily known that the

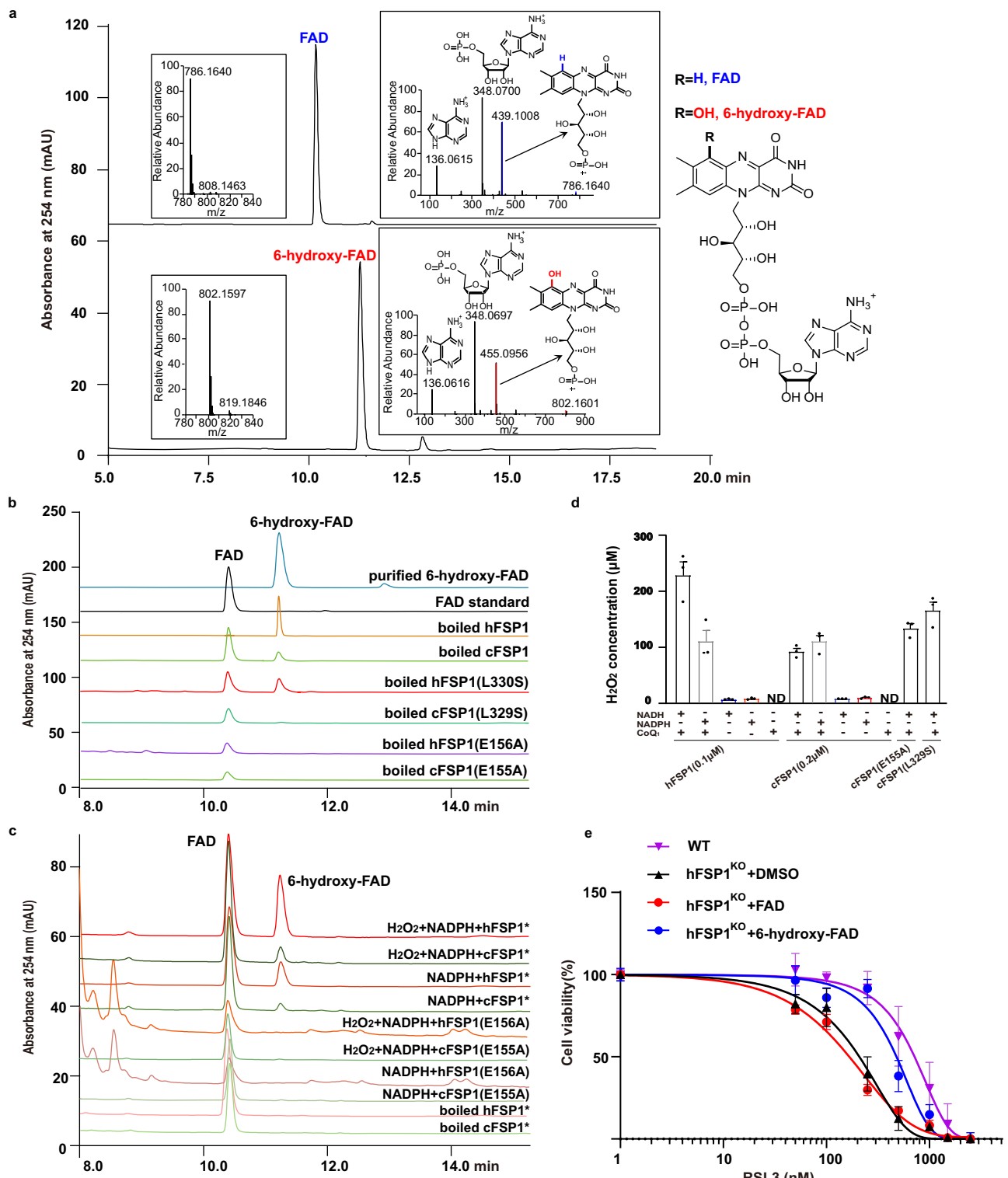

**Fig. 5 | FSP1 also generate 6-hydroxy-FAD to block ferroptosis. a** HPLC-MS/MS chromatogram of the commercial FAD (upper) and the purified 6-hydroxy-FAD (lower). The left insert panels show the MS spectra, the ions at m/z 786.1640 and 802.1597 represent the $[M + H]^+$ ion of FAD and 6-hydroxy-FAD, respectively. The right insert panels show the tandem mass spectra resulting from higher-energy collision dissociation (HCD) of the precursor ions ($[M + H]^+ = 786.16$) and ($[M + H]^+ = 802.16$). The assigned fragments are represented in the spectra, the chemical structure of FAD and 6-hydroxy-FAD are shown in the right part. **b** HPLC analysis of standard FAD (black), purified 6-hydroxy-FAD (blue) and the indicated protein samples. **c** HPLC analysis of the reaction mixtures of the indicated proteins (100 μM) with NADPH or with NADPH and $H_2O_2$, * represent the FAD-reconstituted protein. **d** The indicated FSP1 proteins simultaneously catalyze the reduction of $CoQ_1$ and generate hydrogen peroxide in the presence of NADPH and oxygen, ND, not detected. **e** Dose response of RSL3-induced cell death of HT1080 wild-type (WT) cells and HT1080 hFSP1$^{KO}$ cells treated with DMSO, 10 μM FAD or 6-hydroxy-FAD. Cell viability was assessed using CCK8 assay. The experiments in **a**, **b**, and **c** were performed only once, respectively. Data in **d** and **e** represent the mean ± s.e.m. of three experiments ($n = 3$). Source data are provided as a Source Data file.

modification group at the 6th position should be located in the FAD hydroxylation pocket mentioned above (Fig. 3b, d and Supplementary Fig. 6b), suggesting that the FAD hydroxylation pocket is an active site for FAD hydroxylation and the conserved Glu155 (Glu156 in hFSP1) would be critical. As expected, the recombinant E155A mutation (E156A in hFSP1) did not contain 6-hydroxy-FAD (Fig. 5b) and failed to catalyze the conversion of FSP1-bound FAD to 6-hydroxy-FAD in vitro (Fig. 5c). Meanwhile, the recombinant L329S mutation (L330S mutation in hFSP1) contained both FAD and 6-hydroxy-FAD cofactors (Fig. 5b), indicating that L329S mutation still has a FAD hydroxylation activity. Combined with the NADH oxidation and cell death assays data mentioned above, these results supported that the quinone reduction and FAD hydroxylation occurred at different sites, and both activities should be related to its ferroptosis inhibition activity.

Similar to AIF, FSP1 could generate $H_2O_2$ in NAD(P)H and oxygen-dependent manner, and the $H_2O_2$ amount generated with CoQ was obviously higher than that without CoQ (Fig. 5d and Supplementary Fig. 12), consistent with the NADH oxidation assay result that no obvious NADH consumption was observed in the absence of CoQ (Fig. 2d). Interestingly, the $H_2O_2$ amount generated by E155A or L329S mutations was not less than that of cFSP1 wild type (Fig. 5d), which may be explained by two possible reasons: (i) the $H_2O_2$ amount was a cumulative result of reaction processes, consistent with the NADH oxidation assay result that the two mutations with low NAD(P)H oxidation activity can sustain the entire reaction (Fig. 2d), and (ii) the two mutations did not reduce the enzyme activity of producing $H_2O_2$. Of note, although $CoQ_1$ is required to generate $H_2O_2$ at a low concentration of FSP1, a relatively high concentration of FSP1 (μM level) can produce a large amount of $H_2O_2$ and exhibit an obvious NAD(P)H oxidation activity in the absence of CoQ under aerobic condition (Supplementary Fig. 13). In addition, under the anaerobic condition, FSP1 reduced $CoQ_1$ with a higher $k_{cat}$ compared to those in the aerobic condition (Supplementary Fig. 12 and Supplementary Table 3), indicating that FSP1 can effectively reduce CoQ when only CoQ acts as only electron acceptor substrate. Collectively, these data suggest that both CoQ and oxygen are the substrates of FSP1 proteins and that CoQ may be the preferred substrate and can markedly promote the oxygen reduction to $H_2O_2$, and that oxygen cannot be effectively reduced as only electron acceptor substrate in vitro (Figs. 2d and 5d, Supplementary 12). Furthermore, although $H_2O_2$ is not itself a free radical, it could then react with the isoalloxazine ring of FSP1-bound FAD to generate 6-hydroxy-FAD in FAD hydroxylation pocket through a Glu155-mediated reaction similar to that of benzene hydroxylation by $H_2O_2$ in the presence of catalyst[33].

## 6-hydroxy-FAD inhibits ferroptosis in cells

We next explored whether 6-hydroxy-FAD is directly involved in ferroptosis inhibition. 6-hydroxy-FAD alone can rescue the resistance of HT1080 hFSP1$^{KO}$ cells to RSL3 (Fig. 5e), although its effect on ferroptosis resistance was slightly weaker than that of HT1080 wild-type cells, indicating that 6-hydroxy-FAD can actually act as a new anti-ferroptotic compound to restore ferroptosis resistance in HT1080 hFSP1$^{KO}$ cells. In addition, 6-hydroxy-FAD can also promote the resistance of HT1080 wild-type cells to RSL3 (Supplementary Fig. 14a). Then we used FENIX assay to test whether 6-hydroxy-FAD might be a free radical scavenger. We observed that increasing the concentration of 6-hydroxy-FAD led to further suppression of lipid peroxidation rate (Supplementary Fig. 14b), suggesting that 6-hydroxy-FAD is a potent radical-trapping antioxidant (RTA) in lipid membranes, slightly weaker than another water-soluble RTA BH4[34], and blocks ferroptosis probably by its ability to trap peroxyl radicals. Interestingly, we could not detect 6-hydroxy-FAD in the lysate from HT1080 hFSP1$^{KO}$ or HEK293T cells overexpressing hFSP1 (Supplementary Fig. 15) or HT1080 wild type or HT1080 hFSP1$^{KO}$ cells treated with 6-hydroxy-FAD by LCMS. Therefore, although 6-hydroxy-FAD actually acts a dual role as an active cofactor for FSP1 and a potent antioxidant in vitro and may play the same role in the ferroptosis inhibition in human cells at level below our detection limit, we cannot rule out the possibility that 6-hydroxy-FAD may not be stably stored in human cells and the feeding 6-hydroxy-FAD blocks ferroptosis probably by other unclear mechanism. This should be pursued in the near future.

## The catalytic and anti-ferroptotic mechanisms of FSP1

Like NDH-2 enzymes, FSP1 could form a ternary complex after binding NAD(P)H and ubiquinone at two different sites (Fig. 3d), both of which easily exchange with the cytoplasm and plasma membrane, respectively (Supplementary Figs. 5b and 7c). Our structures reveal that FSP1 contains a unique CTD that mediates the FSP1 functional dimerization (Fig. 2a–c), participates in the formation of two active centers including ubiquinone reduction pocket and FAD hydroxylation pocket (Fig. 3d), and thus play a key role in its oxidoreductase activity and ferroptosis inhibition (Fig. 2d–g). Consistent with previous studies[19–22], we suggest a simple model for the catalytic and ferroptosis suppression mechanism of FSP1 (Fig. 6). In the presence of NAD(P)H and substrates (CoQ or oxygen), FSP1-bound FAD first accepts two electrons from NAD(P)H to form FADH2, which then preferentially transfers two electrons to CoQ, forming reduced CoQ (ubiquinol), or to the oxygen generating $H_2O_2$ at ubiquinone reduction pocket. These two pathways cooperate with each other to promote the NAD(P)H

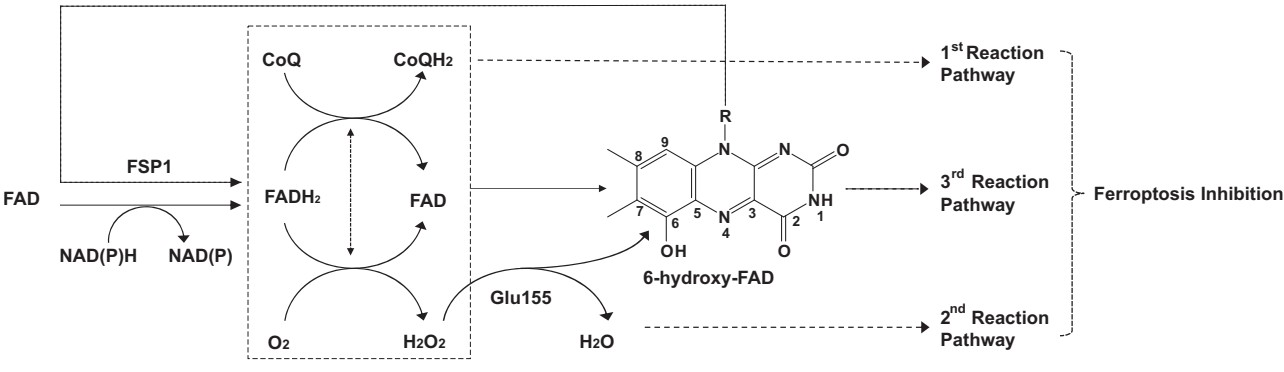

**Fig. 6 | Proposed model for the catalytic and anti-ferroptotic mechanism of FSP1.** FSP1 binds to NAD(P)H and reduces FAD to FADH2, which then transfers two electrons to substrates (such as CoQ and oxygen) at ubiquinone reduction pocket through complex pathways. 1st reaction pathway is to produce the reduced CoQ (ubiquinol) and may be preferred, 2nd reaction pathway is to generate $H_2O_2$ and thereafter react with FSP1-bound FAD to generate 6-hydroxy-FAD through Glu155-mediated reaction at FAD hydroxylation pocket, which is named as 3rd reaction pathway. The 6-hydroxy-FAD can also act as a cofactor to promote the catalytic activity of FSP1. These pathways work together to execute the ferroptosis inhibition of FSP1.

oxidation. In addition, under a possible extensive redox cycling NAD(P)H condition, the FSP1-bound FAD can be hydroxylated to 6-hydroxy-FAD by $H_2O_2$ through Glu155 (Glu156 in hFSP1)-mediated reaction at FAD hydroxylation pocket. Furthermore, 6-hydroxy-FAD can act as a cofactor to further promote the catalytic activity of FSP1. The oxidoreductase activity of FSP1 is intricately regulated by its promiscuity towards the cofactors and substrates such as NAD(P)H, FAD, CoQ, α-tocopherol, $O_2$, $H_2O_2$, and the recently identified vitamin K[17]. Finally, the complex reactions of FSP1 are involved in the regulation of ferroptosis, especially the catalytic products including ubiquinol, reduced hydroquinone (VKH2), and 6-hydroxy-FAD that can suppress the ferroptosis. The model provided by us requires future validation or modification when additional data is available. Nevertheless, our results establish a foundation for the rational design of FSP1 inhibitors that hold promise for developing drugs to treat cancers.

### Limitations of the study

Our study provides a basis to understand how FSP1 reduces CoQ and is involved in anti-ferroptosis. Nevertheless, there are limitations to this study. First, the proteins used in structural determination lack the myristoylation motif, and they may not fully reflect FSP1 conformations associated with the membrane. Second, the reason why CoQ can promote the reduction of oxygen to form $H_2O_2$ was not clearly explained by our data and needs to be elucidated through future studies. Third, although 6-hydroxy-FAD actually acts a dual role as an active cofactor for FSP1 and a potent antioxidant in vitro, its role for FSP1 function and ferroptosis inhibition in human cells needs further research to elucidate.

## Methods

### The cloning, expression, and purification of FSP1

For the heterologous expression in *E. coli*, the gene sequences encoding full-length cFSP1 and hFSP1 were optimized, synthesized (Shanghai Generay Biotech, Co., Ltd., China). The genes of the full-length and truncation mutants of cFSP1 were amplified by standard PCR procedure and subcloned into the pET-22b (Novagen), resulting in a C-terminal 6 × histidine tag. The genes of the full-length and truncation mutants of hFSP1 were subcloned into the pET28a-His-SUMO vector containing an N-terminal 6×his-SUMO tag. Site-directed mutagenesis of FSP1 mutants was performed by PCRs using the QuikChange Site-Directed Mutagenesis Kit. All gene sequences were confirmed by DNA sequencing.

FSP1 proteins were overexpressed in *E. coli* BL21 (DE3) cells that were grown in LB medium at 37 °C to an OD600 of 0.7–0.8 before being induced with 0.1 mM IPTG at 18 °C for 16 h. The cells were collected by centrifugation, resuspended in the phosphate buffer saline (PBS) pH 7.4 supplemented with 150 mM KCl, 5% (v/v) glycerol, and 5 mM β-mercaptoethanol (buffer A), and then lysed using a mini ultra-high pressure cell disrupter (JNBIO, China), followed by centrifugation at 12,000 × *g* for 1 h. Lysate supernatant was applied to Ni-Chelating Sepharose Fast Flow resin (IMAC, GE Healthcare, USA) at 4 °C. The resin was washed with buffer A supplemented with 20 mM imidazole, and the bound FSP1 proteins were eluted from the affinity resin with buffer A supplemented with 250 mM imidazole. The eluted cFSP1 proteins were concentrated and then further purified by Superdex 200 Increase 10/300 GL columns (GE Healthcare, USA) with buffer B containing 20 mM Tris-HCl pH8.0, 150 mM NaCl. The eluted hFSP1 proteins were treated with Ulp1 to cleave off the 6×his-SUMO tag, dialyzed against buffer A, re-loaded onto the Ni-Chelating Sepharose Fast Flow resin, and then the flow-through was concentrated and further purified by Superdex 200 Increase 10/300 GL columns (GE Healthcare, USA) with buffer B.

To render cFSP1$^{\Delta N}$ more amenable to crystallization, we performed a reductive methylation to modify surface lysine residues by using a protocol derived from a previous study[23]. Briefly, cFSP1$^{\Delta N}$

eluted from the Ni-chelating resin was diluted with 0.1 X PBS buffer supplemented with 5% (v/v) glycerol and 5 mM β-mercaptoethanol and loaded onto a HiTrap Heparin HP column (GE Healthcare, USA). The bound protein was eluted with a linear gradient of PBS buffer supplemented with 1 M KCl buffer and then modified by reductive methylation with formaldehyde and Dimethylamine–borane complex. Subsequently, the methylated cFSP1$^{\Delta N}$ was concentrated and then further purified by Superdex 200 Increase 10/300 GL columns with buffer B. For phase determination of the cFSP1$^{\Delta N}$ crystal structure, selenomethionie (Se-Met)-derivatized cFSP1$^{\Delta N}$ was expressed by bacterial growing in M9 medium supplemented with 100 mg/L L-selenomethionine. The methylation and purification procedure of SeMet-derivated protein was the same as mentioned above for the methylated cFSP1$^{\Delta N}$.

### Crystallization, data collection, and structure determination

Extensive crystallization trials were performed for FSP1 proteins, most of the efforts filed to yield crystals or diffraction-quality crystals. Finally, the methylated cFSP1$^{\Delta N}$ was concentrated to ~3 mg/ml and gave rise to diffraction-quality crystals by sitting-drop vapor diffusion method at 20 °C, using equal volumes of proteins and well solution. The crystals of cFSP1$^{\Delta N}$ and SeMet labeled cFSP1$^{\Delta N}$ were grown from the well solution containing 1.8 M $(NH_4)_2SO_4$, 0.1 M Bis-Tris pH6.5 and 3% (w/v) PEG550MME. The crystals of cFSP1$^{\Delta N}$ and SeMet labeled cFSP1$^{\Delta N}$ were flash-frozen in liquid nitrogen with the well solution plus 10% (v/v) glycerol as cryoprotectant. For the crystallization of the cFSP1$^{\Delta N}$-CoQ$_1$ complex, the methylated cFSP1$^{\Delta N}$ was pre-mixed with 300 μM CoQ$_1$, the crystals of the complex were grown from the well solution containing 14% w/v PEG8000, 0.1 M MES pH6.2, 160 mM Calcium acetate and 20% v/v glycerol, and then flash-frozen in liquid nitrogen with a cryoprotectant consisting of 30% w/v PEG8000, 0.1 M MES pH6.0, 160 mM Calcium acetate, 5% v/v glycerol and 2 mM CoQ$_1$.

All diffraction datasets were collected from one crystal each on beamline BL19U1 or BL17U1 at Shanghai Synchrotron Radiation Facility (Shanghai, China) using a charge-coupled device (CCD) detector and processed by using the XDS data processing package[35,36] or the HKL3000 package[37]. Further processing was carried out with programs from the CCP4 suite[38]. The structure of cFSP1$^{\Delta N}$ was determined by the single-wavelength anomalous dispersion (SAD) method using the PHENIX package[39]. A crude model was automatically traced using the program RESOLVE[40]. Manual model building and refinement were performed iteratively with the program COOT[41] and PHENIX. The structure of cFSP1$^{\Delta N}$-CoQ$_1$ complex was determined by molecular replacement with PHASER[42] using the cFSP1$^{\Delta N}$ structure as a search model. Manual model building and refinement were also performed iteratively with the program COOT and PHENIX. The final models were validated using Molprobity[43], with the final statistics for data collection, structure determination, and refinement given in Supplementary Table 1. Structural presentations were prepared using Pymol (https://pymol.org/2) or ChimeraX[44].

### NAD(P)H oxidation and enzyme kinetics assays

NAD(P)H oxidation activity and enzyme kinetics of FSP1 proteins were measured by monitoring the decrease in NAD(P)H absorbance at 340 nm with a microplate spectrophotometer (AMR-100, Allsheng, Hangzhou, China). For NAD(P) oxidation assays in aerobic or anaerobic conditions, the reaction mixture in a final volume of 100 μl contained 50 mM Tris-HCl (pH 8.0), 250 mM NaCl, 500 μM NAD(P)H, 200 μM CoQ$_1$ and 0.5 μM of recombinant cFSP1 proteins or 0.2 μM of recombinant hFSP1 proteins (other concentrations are specified). For anaerobic conditions, the reactions were performed in a $N_2$ filled glove box (Mikrouna, China, Universal 2440/750/900, $O_2$ content <200 ppm), and all the sample solution was degassing under $N_2$ before use. For enzyme kinetic assays, the reactions contained 50 mM Tris-HCl (pH 8.0), 250 mM NaCl, FSP1 proteins at a constant final concentration

(50 nM cFSP1, 25 nM hFSP1, 25 nM 6-hydroxy-FAD-reconstituted hFSP1 or 50 nM FAD-reconstituted hFSP1 in aerobic condition, or 10 nM hFSP1 in anaerobic condition), either with 400 μM $CoQ_1$ and varied NAD(P)H concentration (to determine kinetics for NAD(P)H) or with 500 μM NAD(P)H and varied $CoQ_1$ concentration (to determine kinetics for NAD(P)H). Multiple reactions were started by the addition of the substrate $CoQ_1$ and carried out by using a 96-well plate, and measurements were made at one-minute intervals over a time period in aerobic or anaerobic conditions. The curves, which were generated using GraphPad Prism 9 (GraphPad Software Inc., USA) and shown in figures, display the average result of three experiments. In order to simply compare the NAD(P)H oxidation activity of the wild type and mutants of cFSP1 or hFSP1, we chose to compare the initial activity that was derived from the decrease in the absorbance at 340 nm within five minutes of the reaction. In order to determine the enzyme kinetics, we chose the enzyme rate that was drived from the decrease in the absorbance at 340 nm within 10 min of the reaction, then the enzyme kinetics were calculated by constructing a Lineweaver-Burk plot.Each assay was repeated three times.

### Human cell lines and hFSP1 expression in human cells

Human fibrosarcoma (HT1080, Cat. # TCHu170) and human embryonic kidney 293 T (HEK293T, Cat. # GNHu17) cells were obtained from the Cell Bank of Shanghai Institute of Biochemistry and Cell Biology (Chinese Academy of Sciences, Shanghai, China), and regularly tested for mycoplasma contamination. HT1080 $hFSP1^{KO}$ cells were generated by using CRISPR/Cas9 system-based technology derived from previous publication[3]. Briefly, Single sgRNA guides (CACCGCACTCTCA TTCACTCCCAAG, AACCTTGGGAGTGAATGAGAGTGC) were designed to target critical exons of the genes to be inactivated. Guides were cloned into LentiCRISPRv2 vector (Addgene #52961), and then the plasmid was transfected into HT-1080 cell using the Lipofectamine 3000 reagent according to the manufacturer's recommendations (ThermoFisher scientific). After 48 h incubation, the cells were selected with puromycin (1 μg ml⁻¹) for 3 days. FSP1-knockout single clones were then screened and acquired after continuing to culture for 2–3 weeks without selective antibiotics. Western blot analysis was used to confirm successful gene deletion. All human cells were cultured in a 37 °C incubator with 5% $CO_2$, HT1080 and HT1080 $hFSP1^{KO}$ cells were maintained in Dulbecoo's modified Eagle's medium (DMEM) supplemented with 10% fetal bovine serum and 1% Penicillin/Streptomycin solution, HEK293T cells were maintained in DMEM with 4.5 g/L Glucose and GlutaMAX™ (Gibco).

For wild-type or mutant hFSP1 expression in HEK293T or HT1080 $hFSP1^{KO}$ cells, total RNA was extracted from HT1080 cells by using TRIzol Reagent (Ambion), then reverse transcription was performed to generate cDNA from total RNA by using HiScript II Q RT SuperMix for qPCR (+gDNA wiper) (Vazyme, China), then the genes of the full-length and truncation mutants of hFSP1 was amplified from the synthesized cDNA by standard PCR procedure, and subcloned into the pcDNA3.1 (Invitrogen). The E156A and L330S mutants of hFSP1 were subsequently generated using site-directed mutagenesis. All gene sequences were confirmed by DNA sequencing. The plasmids were transfected into the HEK293T or HT1080 $hFSP1^{KO}$ cells using Lipofectamine™ 3000 transfection Reagent (Thermo Fisher). Fresh medium were added into culture plates after 5 hours transfection, and cells were treated differently (for details, see the following sections) after 36–48 h of transfection.

### Human cell fractionation, western blot, and co-IP

The transfected cells were harvested and washed with PBS. For HPLC-MS analysis, the cell pellet was resuspended in 200 μl of water, lysed by heating at 85 °C for 5 min, and centrifuged at 13000 x g for 25 min, and then the supernatant was subjected to HPLC-MS analysis. For western blot, the cell pellet was resuspended in RIPA lysis buffer (Beyotime,

China) supplemented with 1% protease inhibitor cocktail (Sigma-Aldrich), homogenized by vortex, and centrifuged at 13000 x g for 30 min. The protein concentration of the supernatant was determined by a Bradford protein assay kit (Beyotime). 20 μg proteins from each sample were separated by 10% SDS-PAGE, transferred onto nitro-cellulose filter membrane (Millipore Corp.), and blocked with 5% fat-free milk for 1 h at room temperature. The primary antibodies against FSP1 (1:1000; Proteintech, Cat. #20886-1-AP) were incubated overnight at 4 °C, sequentially the Peroxidase-Affinipure goat anti-rabbit IgG (H + L) (1:3000; Jackson ImmunoResearch, code 111-035-003) and enhanced chemiluminescence solution (GE Healthcare) were used for visualizing protein expression. With the same protocol, the primary antibodies against HSP90 (1:2000;ZSGB-BIO, Cat. #TA-12) or againstβ-actin (1:2000;ZSGB-BIO, Cat. #TA-09) were incubated and followed by Peroxidase AffiniPure Goat Anti-mouse IgG (H + L) (1:3000; Jackson ImmunoResearch Laboratories, Code#115-035-003).

For co-IP experiments, HEK239T cells were transfected with an equal amount of vectors (pcDNA3.1-hFSP1-Myc, pLenti-hFSP1-Flag, and empty pLenti). After 36 h, cell pellets were lysed in BC100 lysis buffer supplemented with 1% protease inhibitor cocktail (Sigma-Aldrich), homogenized by vortex, and centrifuged at 4 °C. 50 μL supernatant from each sample was saved as input controls, and all remaining supernatant was incubated with 20 μL anti-FLAG M2 Affinity Gel (Sigma-Aldrich) overnight at 4 °C. The Gel was washed 3–6 times with lysis buffer, and the bound proteins were eluted in SDS buffer and analysed by western blot as described above. Anti-FLAG (Invitrogen, Cat. #PA1-984B) or anti-Myc (CST, Cat.#2278) antibodies were used at a dilution of 1: 1000 for immunoblotting.

### Human cell death and viability assays

For cell death assay, the transfected cells were harvested and seeded in 24-well plates at a density of 100000 per well, then treated with DMSO, RSL3 (75 nM), or with RSL3 (75 nM) and Ferrostatin-1 (Fer-1, 2 μM). After 3 h treatment, cells were stained by Sytox Green (Thermo Fisher) for 15 min. The relative cell death was recorded by fluorescence microscope and analyzed by Graphpad Prism 8 software. For cell viability assay, HT1080 and HT1080 $hFSP1^{KO}$ cells were seeded in 96-well plates at a density of 6000 per well, respectively. After 18 h of seeding, the cells were treated with RSL3 (final concentration ranged from 50 nM to 2.5 μM) and other compounds (0.1% dimethyl sulfoxide [DMSO], 10 μM FAD or different concentrations of 6-Hydroxy-FAD). After 24 h of treatment, cell counting kit-8 (CCK-8, Dojindo Laboratories, Japan) was added into each well and incubated with cells for 30 min. Absorbance at 450 nm was measured using a microplate reader. The relative cell viability was normalized to that of the DMSO-treated cells and analyzed by GraphPad Prism 8 software.

### FENIX assays

The fluorescence-enabled inhibited autoxidation (FENIX) assay was used to determine the lipid radical-trapping activities of chemicals[45]. Egg PC liposomes (extruded to 100 nm, 1 mM), STY-BODIPY (1 μM), and indicated concentration of liproxstatin-1, $BH_4$, 6-Hydroxy-FAD or vehicle (DMSO) were vortexed in PBS (10 mM, pH7.4), then 200 μl aliquots of liposomes were incubated in 96-well plates at 37 °C for 20 min. Then, DTUN (200 mM in EtOH) were added to the aliquots. The plate was mixed for 5 min and kinetic data of STY-$BODIPY_{OX}$ was acquired at 485 nm (λex) and 528 nm (λem) by Mithras LB940 micro-plate reader (Berthold Technologies). Of note, FAD is not suitable for the FENIX assay because FAD has a similar fluorescence signal (λex = 460, λem = 550) to STY-$BODIPY_{OX}$.

### Gel-filtration assays

FSP1 proteins purified as described above were subjected to gel-filtration analysis (Superdex 200 Increase 10/300 GL columns; GE Healthcare) separately. The assays were performed in buffer B with a

flow rate of 0.5 ml min$^{-1}$ and an injection volume of 0.1 ml protein sample at 20 °C. The gel filtration molecular weight markers (Cat. MWGF1000-1KT, Sigma-Aldrich) were used to estimate the apparent molecular weight of FSP1 proteins in solution. The FSP1 proteins were visualized by SDS-PAGE followed by Coomassie blue staining.

## Dynamic light scattering (DLS) assay
DLS measurements were performed at 8 °C or 20 °C using a DynaPro NanoStar instrument (Wyatt Technology Corporation) with disposable cuvettes. The protein concentrations were 5–10 μM in buffer B and cleared by centrifugation at 13,000 × $g$ and 4 °C for 20 min. Measurements were carried out in triplicate, with 10 cycles per measurement, 5 s in each cycle. The analysis was performed with the Dynamics 7.0 software using the regularization method (Wyatt Technology). We determined hydrodynamic radius, and polydispersity, and calculated molecular weight.

## Preparation of 6-hydroxy-FAD
6-hydroxy-FAD was purified from the recombinant hFSP1. In brief, hFSP1 purified from Ni-Chelating resin was firstly denatured by heating at 85 °C for 10 min, followed by centrifugation at 13,000 × $g$ for 20 min. The supernatant was then adjusted to pH 4.0 with acetic acid, loaded onto a HiTrap Q HP column that was installed on FPLC ÄKTA purifier system (GE Healthcare) and pre-equilibrated in pH 4.0 water, and eluted with a linear gradient of pH 4.0 water supplemented with 500 mM NaCl. Subsequently, the peak fraction corresponding to the target 6-hydroxy-FAD was collected, supplemented with methanol at a final concentration of 5% (v/v) and acetic acid at a final concentration of 1% (v/v), loaded onto a supelclean™ LC-18 SPE column (Sigma-Aldrich), washed with a solution containing 5% methanol and 1% acetic acid, and water in that order, and then eluted by using 100% methanol. Finally, the elution was freeze-dried by a vacuum-freezing dryer (Heto-Holten, Denmark) to yield a dark green powder. The 1H NMR spectra were recorded on a Bruker Avance DRX-600 spectrometer (Bruker Corp., Switzerland) in D$_2$O (with TMS as the internal standard). Chemical shifts ($\delta$) are expressed in ppm relative to the solvent signals.

## HPLC and HPLC-MS/MS
HPLC analysis was performed using two Shimadzu LC-10AT VP pumps gradient-controlled HPLC system, equipped with photodiode array detector (SPD-10AVP), attached with Shimadzu LC Solution program. High resolution HPLC-MS/MS was performed on a rapid separation liquid chromatography system (Dionex, UltiMate3000, UHPLC) coupled with an ESI-Q-TOF mass spectrometer (Bruker Daltonics, Impact HD). The MS analysis was performed under the following conditions: ESI-positive mode for scanning, ESI capillary voltage with 4.5 kV, nebulizer gas nitrogen with 0.5 bar, dry gas nitrogen with 6 L/min, probe temperature with 200 °C, full scan mass range from 50 to 1500 m/z. The precursor ions intensity threshold greater than 3.0E3 in the quadrupole were selected for auto MS-MS fragmentation analysis. Exclusion was used after 3 spectra within 60 s. The data were analyzed using Bruker Data Analysis software. Samples were separated by a YMC-pack pro-C18 reverse phase column (4.6 × 250 mm, 5 μm, YMC, Japan) at UV 254 nm. Phase A contained 5 mM ammonium acetate (pH 5.0) and Phase B contained 100% acetonitrile. The sample was eluted using a linear gradient from 5% B to 50% B at a flow rate of 1 ml/min over 15 min. 13 samples used for LC-MS/MS analysis included FAD standard (Sigma-Aldrich, Cat. # F6625), the purified 6-hydroxy-FAD, boiled purified hFSP1, boiled purified cFSP1 proteins, the lysates from *E. coli* BL21(DE3) with empty vector or expressing hFSP1, expressing cFSP1, HT1080 hFSP1$^{KO}$ cells or with empty vector, expressing hFSP1, HEK293T cells or with empty vector, expressing hFSP1. The LC-MS/MS experiments of each sample were performed only once.

## Micro-Fourier transformed infrared (micro-FTIR) analysis
Micro-FTIR spectra were collected on a Thermo-Fisher Nicolet iN10 MX FTIR spectrometer equipped with an IR microscope. All the spectra were recorded under the transmission mode with a resolution of 4 cm$^{-1}$ and averaged for 16 scans.

## Preparation of FAD-reconstituted cFSP1, hFSP1 and 6-hydroxy-FAD-reconstituted hFSP1
The reconstitution of FSP1 was performed by using a protocol derived from a previous publication[20]. In detail, cFSP1 or hFSP1 from cell lysate supernatant was bound to Ni-Chelating resin in buffer A, and washed with buffer A supplemented with 20 mM imidazole, and an excessive amount of solution containing 2 M potassium bromate and 0.5 M urea until cofactors removal was complete as judged by complete loss of green color and by HPLC. Next, the cFSP1 or hFSP1-bound resin was washed with 300 mL of buffer A, and then incubated with buffer A supplemented with 500 μM FAD or 6-hydroxy-FAD prepared by ourselves at 4 °C for 3 h, followed by extensive washing with buffer A to remove all unbound cofactor. Subsequently, FAD-reconstituted cFSP1, hFSP1, or 6-hydroxy-FAD-reconstituted hFSP1 was eluted from the resin with buffer A supplemented with 250 mM imidazole, dialyzed against a buffer containing 20 mm Tris pH 8.0 and 300 mM KCl, and then frozen at - 80 °C for further study. In order to test whether FAD-reconstituted FSP1 can catalyze the conversion of FAD to 6-hydroxy-FAD, we incubated FAD-reconstituted cFSP1 or hFSP1 (100 μM) with 10 mM NADPH and with or without 25 μM H$_2$O$_2$ at room temperature for 30 min. Subsequently, these reactions were terminated by heating at 85 °C for 5 min, then centrifuged and analyzed by HPLC.

## Measurement of hydrogen peroxide production during enzyme reaction of FSP1
The content of H$_2$O$_2$ during in vitro enzyme reaction of FSP1 was determined in aerobic or anaerobic conditions by using a hydrogen peroxide fluorometric assay kit (Elabscience, China), which takes advantage of H$_2$O$_2$ reacting with the fluorescent probe in the presence of peroxidase to generate fluorescence intensity ($\lambda_{ex}$ = 535 / $\lambda_{em}$ = 587 nm) that is proportional to the H$_2$O$_2$ concentration range from 0.02 to 10 μM. Measurements were performed with the fluorometric assay in 96-well black bottom plate according to the manufacturer's protocol. Briefly, in order to qualitatively test whether FSP1 generates H$_2$O$_2$, reaction mixtures containing 500 μM NAD(P)H, FSP1 (2 μM cFSP1 or hFSP1, or no protein) and CoQ$_1$ (0 or 200 μM) in buffer C (50 mM Tris-HCl pH 8.0, 250 mM NaCl) were incubated at room temperature for 20 min. Subsequently, 25 μL of each reaction mixture for qualitative analysis was mixed with 25 μL of reagent 1 (buffer solution), and added into the well of 96-well plate, followed by the addition 50 μL of working solution. After gently shaking for 10 s, the plate was incubated in the dark at room temperature for 10 min. Then the fluorescence intensities ($\lambda_{ex}$ = 535 / $\lambda_{em}$ = 587 nm) were measured and recorded by Enspire multimode plate reader (PerkinElmer, USA) at 1 min intervals for 30 cycles. In order to detect the content of H$_2$O$_2$ during the enzyme reaction, reaction mixtures containing NAD(P)H (0 or 500 μM), FSP1 (0.1 or 0.16 μM hFSP1, 0.2 μM cFSP1, or no protein) and CoQ$_1$ (0 or 200 μM) in buffer C were incubated at room temperature for 60 min, and then 1 μL of each reaction mixture for quantitative analysis was mixed with 49 μL of reagent 1. Subsequently, 50 μL of each diluted reaction mixture was transferred into the well, mixed with 50 μL of working solution, incubated, and measured as above. The quantitative production of H$_2$O$_2$ by FSP1 was calculated using a standard fluorescent curve prepared with known concentrations of H$_2$O$_2$.

## Statistics and reproducibility

Apart from the co-IP and SDS-PAGE experiments, which were performed only once, all the western blots, images and DLS are representative of at least two independent experiments with similar results. Apart from FENIX, Gel-filtration assays, micro-FTIR, NMR, HPLC, and HPLC-MS analysis, which were performed only once, all the values were expressed as the mean ± standard error of the mean of three experiments ($n = 3$). The difference between the two groups was analyzed by two-sided one-way ANOVA with Tukey's test. Statistical analysis was carried out with GraphPad Prism 8.0 or 9.0 software. All statistical significances were set as $p \leq 0.05$.

## Reporting summary

Further information on research design is available in the Nature Portfolio Reporting Summary linked to this article.

## Data availability

The data that support this study are available from the corresponding author upon request. Coordinates and structure factors files have been deposited in the Protein Data Bank with the accession codes 7XPI (cFSP1$^{\Delta N}$) and 7YTL (cFSP1$^{\Delta N}$-CoQ$_1$ complex). All data generated or analyzed during this study are included in this article and its Supplementary information file. Source data for the Figures, Supplementary Tables, and Figures are provided as a Source Data file. The protein structures used for analysis in the study are available in the Protein Data Bank under accession codes 4NWZ, 4G6G, 1M6I, and 5KMS. The HPLC-MS datasets are available from figshare (https://doi.org/10.6084/m9.figshare.23255453)[46]. Source data are provided with this paper.

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

## Acknowledgements

We thank Hongxiang Lou, Yuemao Shen, and Shuangjiang Liu (Shandong University) for their support, the staff at Shanghai Synchrotron Radiation Facility beamline 19U and 17U for assistance with data collection, Xinwei Sun, Minzhi Jiang, Chengjia Zhang, Nannan Dong (State Key Laboratory of Microbial Technology, Shandong University) for assistance in experiments under anaerobic condition, Zhifeng Li, Jingyao Qu, Guannan Lin and Xiangmei Ren from the Core Facilities for Life and Environmental Sciences, State Key Laboratory of Microbial Technology, Shandong University for the assistance in LC-MS, DLS, and plate reader analysis. This work was supported by funds from the Shandong Province Natural Science Foundation grants ZR2020MC051 to D.Z., the Hubei Province Natural Science Foundation grants 2015ZFB763 to Q.W., the Key R & D Project of Shandong Province grants 2019GSF108212 and the Fundamental Research Funds of Shandong University grants 2018JC004 to D.Z., Core Facility Improvement Funds of Shandong University grants ts20220208 to J.Z., and the National Natural Science Foundation of China grants 32201085 to C.L.

## Author contributions

D. Zhu initiated and designed the overall project. D. Zhu and H. Yuan supervised the overall project. B. Chu supervised the study of the anti-ferroptotic function of FSP1. D. Zhu and Q. Wang supervised the determination of the two crystal structures. Y. Lv, Q. Sun, and D. Zhu grew and optimized the crystals, and collected the X-ray diffraction data. Y. Lv, D. Zhu, and Q. Wang determined the structure. Y. Lv, Q. Sun, J. Zhu, H. Xu, X. Li, Y. Li, and D. Zhu performed other in vitro experiments. C. Liang performed the experiments in human cells. Y. Lv, Q. Wang, and D. Zhu wrote the manuscript. All authors read and edited the manuscript.

## Competing interests

D. Zhu and Y. Lv are filing a China patent application (202310139279.0) based on the results reported in this paper. All other authors declare no competing interests.
