## [Peer Review File · Nature Communications]

REVIEWER COMMENTS

Reviewer #1 (Remarks to the Author):

Ferroptosis suppressor protein 1 (FSP1) prevents lipid peroxidation and ferroptosis through its role as an oxidoreductase, exploiting NAD(P)H to generate the reduced form of coenzyme Q10 (i.e., ubiquinol) which acts as a radical trapping antioxidant to break the lipid peroxidation chain reaction. FSP1 has been shown to employ a variety of other substrates, such as vitamin E and vitamin K. No structural information for FSP1 is available, though it has been proposed to have a similar structure and mechanism as other well-known CoQ oxidoreductases such as AIF and NQO1 based on homology and predicted structures (supported by some biochemical analysis of mutations).

Here, the authors solve the first crystal structures of chicken FSP1, lacking the first 11 amino acids involved in myristoylation and membrane anchoring. A structure is also obtained for FSP1 bound to FAD and its substrate CoQ1. The structures reveal the expected oxidoreductase structure with a classic Rossmann fold. In the crystal structures, FSP1 is present as a dimer mediated by interactions between the c-terminal domain (CTD). Support for the dimer structure is provided by HPLC and dynamic light scattering, though there are some differences between the human and chicken FSP1 proteins. Based on cell and in vitro analysis of a CTD deletion mutant of FSP1, the authors suggest that the FSP1 dimer is physiologically relevant. The authors also discover that FSP1 can catalyze the production of H₂O₂ in vitro, independent of CoQ, and 6-hydroxy-FAD. They further suggest that 6-hydroxy-FAD increases FSP1 oxidoreductase activity and that it may also be released from FSP1 and have the ability to directly suppress ferroptosis.

The structure of FSP1 is important. The dimerization of FSP1 and generation of H₂O₂ and 6-hydroxy-FAD are potentially interesting. However, several of the conclusions remain speculative and are not fully supported. Additional data is required to support the proposed models and rule out other possibilities.

Major Comments

1) Although the dimeric structure is potentially interesting, additional characterization is required to support the conclusion that this is physiologically relevant.

- In the gel filtration chromatography data (Figure S3A), the hFSP1 Δ CTD runs at a larger size than hFSP1. This does not make sense. The authors argue that hFSP1 only dimerizes under conditions of high NADH and that this dimerization is CTD dependent, but the gel filtration is not consistent with this claim.

- Analysis of the Δ CTD mutant by dynamic light scattering in the presence and absence of NADH would be useful to support the dimerization conclusions.

- hFSP1 is active at much lower NADH concentrations than are required for induced dimerization. Doesn't this argue that the dimerization is not required for its activity?

- No data is provided to demonstrate that FSP1 dimerizes in cells. While the Δ CTD mutant does not rescue ferroptosis, this may be due to effects on dimerization or on its coordination of CoQ and its oxidoreductase activity. Does FSP1 dimerize in cells? This could be addressed by several approaches, such as FRET or CoIP.

- Can the authors comment on previous studies showing that C-terminally tagged GFP FSP1 is functional and able to suppress ferroptosis? Would this large C-terminal tag be expected to disrupt the putative dimer and if so would this suggest that the dimer is not necessary?

2) To support the conclusion that hFSP1 oxidizes NADH and NADPH with similar efficiency (line 261), full enzyme kinetics should be performed (i.e., Michaelis Menten kinetics). Similarly, enzyme kinetics should also be performed comparing the 6-hydroxy-FAD and FAD hFSP1 to support the conclusions (lines 340-342).

3) It is absolutely required that peaks are assigned in MS/MS spectra. In Figure 4D, assign fragments to the MS/MS spectrum and show a comparison with pure FAD to rule out that FAD did not just form a water adduct during ionization. In Figure S8, assign peaks to the carbon atoms in the structure and compare to native FAD.

4) It is mentioned isomer forms of FAD (hydroxyl v ketonyl) could not unambiguously distinguished. Could this be distinguished by IR spectroscopy (rule out carbonyl species)? This is an important point to support the conclusions related to 6-hydroxy-FAD.

5) It is stated that "the amount of the generated H₂O₂ with CoQ was obviously higher than that without CoQ, which is small and as a cumulative result over the whole analysis period (Fig. 4G), consistent with the NADH oxidation assay result that no obvious NADH 370 consumption was observed in the absence of CoQ". It is unclear why CoQ required to generate H₂O₂? In this scenario CoQ is present and FSP1 would employ it as the preferred substrate over oxygen. Unclear under which conditions CoQ vs oxygen would be employed. Enzyme kinetics could be useful to support the conclusions.

6) Based upon data using DPPH, the authors conclude that 6-hydroxy-FAD is not a free radical scavenger. DPPH is not a radical trap / reporter. This is not the correct experiment to test whether it scavenges lipid peroxy radicals. See Shah et al., Cell Chem Biol 2019 (<https://doi.org/10.1016/j.chembiol.2019.09.007>). This reference (which is cited) makes a point that DPPH should not be used to derive / measure lipid radical scavenging activities. All DPPH assays should be reevaluated, and other methods employed.

Minor Comments

- 1) A brief statement of why chicken FSP1 was analyzed vs human FSP1 would be helpful.
- 2) Figure 2D, the overlaid curves are not easily distinguished. Please use different colors / shapes.
- 3) Figure 2G, western blot is of unacceptable quality.
- 4) Figure 4E, use an offset of the chromatograms and consider quantifying the peaks. As shown, it is very difficult to compare relative peak intensities.

Reviewer #2 (Remarks to the Author):

FSP1 is a recently discovered glutathione-independent ferroptosis suppressor, but its underlying structural mechanism has remained unknown. In this manuscript, the authors report the crystal structure of chicken (*Gallus gallus*) FSP1 (cFSP1) in the substrate-free and ubiquinone-bound states. The authors found that FSP1 has a FAD-binding domain, a NAD(P)H-binding domain, and a unique C-terminal domain. The C-terminal domain mediates functional dimerization of FSP1 and participates in the active site formation. FSP1 catalyzes the formation of 6-hydroxy-FAD, and the 6-hydroxy-FAD containing cFSP1 is catalytically active. Importantly, in the absence of FSP1, 6-hydroxy-FAD, but not FAD, can rescue ferroptosis. Therefore, this study establishes 6-hydroxy-FAD as an active cofactor of FSP1 and a potent ferroptosis inhibitor. Overall, these are significant findings that would merit the publication in *Nature Communications*, providing the following issues are adequately addressed.

Major:

How 6-hydroxy-FAD inhibits ferroptosis needs to be better explained. Currently, this molecule is not connected with any known ferroptosis resistance mechanism. How does this molecule inhibit ferroptosis in the absence of FSP1?

Can the authors capture 6-hydroxy-FAD in their crystal structure? Since the authors can isolate 6-hydroxy-FAD, this is worth a shot, as this would provide strong support for the proposed mechanism.

Minor:

(line 121): “though it is partially disordered and displayed weak density” should be “though it is partially disordered and displays weak density”.

Reviewer #3 (Remarks to the Author):

In the present work Lv et al., report on the crystal structure of FSP1 (previously known as AIFM2) unbound and bound to one of its substrates, ubiquinone. The study appears to be carefully conducted and the report of the FSP1 structure is timely and of interest as it could catalyse a deeper understanding of the biology of FSP1 as well as helping to understand the mode of actions of FSP1 inhibitors.

The current revision does deep into the specifics of the structural work as I dont feel capable to judge these aspects. Therefore I have only limited my comments to particular aspect of the biology of FSP1 and ferroptosis.

One of aspect that I find not entirely convincing is regarding the role of 6-hidroxyFAD (6OH-FAD) in ferroptosis. Based on the observation that 6OH-FAD is formed in a H₂O₂ dependent manner during catalysis in vitro, the authors posit that this intermediate could be formed in cells and could is involved in protecting cells from ferroptosis. While in Figure 4H they show that 6OH-FAD can suppress RSL3 induced ferroptosis this might not be entirely surprising given the presence of the 6-OH group that could work as a direct antioxidant (the DPPH assay provided in SI is not sufficient to exclude this). Still, I find remarkable that 6OH-FAD can be directly taken up by cells; to the best of my knowledge FAD is, if at all, very poorly taken up by cells. The question remains, if cells can produce 6OH-FAD at sufficient high levels to make it a relevant inhibitor. Did the authors attempt to measure it in cells and whats the contribution of FSP1 to its levels? I dont think this disqualifies the work but some discussion and potentially toning down some oft he conclusions seems justifiable.

Responses in Point-by-Point to the Comments of Referees

Reviewer #1:

Ferroptosis suppressor protein 1 (FSP1) prevents lipid peroxidation and ferroptosis through its role as an oxidoreductase, exploiting NAD(P)H to generate the reduced form of coenzyme Q10 (i.e., ubiquinol) which acts as a radical trapping antioxidant to break the lipid peroxidation chain reaction. FSP1 has been shown to employ a variety of other substrates, such as vitamin E and vitamin K. No structural information for FSP1 is available, though it has been proposed to have a similar structure and mechanism as other well-known CoQ oxidoreductases such as AIF and NQO1 based on homology and predicted structures (supported by some biochemical analysis of mutations).

Here, the authors solve the first crystal structures of chicken FSP1, lacking the first 11 amino acids involved in myristylation and membrane anchoring. A structure is also obtained for FSP1 bound to FAD and its substrate CoQ1. The structures reveal the expected oxidoreductase structure with a classic Rossman fold. In the crystal structures, FSP1 is present as a dimer mediated by interactions between the c-terminal domain (CTD). Support for the dimer structure is provided by HPLC and dynamic light scattering, though there are some differences between the human and chicken FSP1 proteins. Based on cell and in vitro analysis of a CTD deletion mutant of FSP1, the authors suggest that the FSP1 dimer is physiologically relevant. The authors also discover that FSP1 can catalyze the production of H₂O₂ in vitro, independent of CoQ, and 6-hydroxy-FAD. They further suggest that 6-hydroxy-FAD increases FSP1 oxidoreductase activity and that it may also be released from FSP1 and have the ability to directly suppress ferroptosis.

The structure of FSP1 is important. The dimerization of FSP1 and generation of H₂O₂ and 6-hydroxy-FAD are potentially interesting. However, several of the conclusions remain speculative and are not fully supported. Additional data is required to support the proposed models and rule out other possibilities.

Response: Thanks for the positive comments. We have performed the additional experiments to support the conclusions. The dynamic light scattering of FSP1 proteins including cFSP1^{ΔCTD} and hFSP1^{ΔCTD} at two different temperature condition (8 °C or 25 °C) (Supplementary Fig. 3), and Co-IP experiments (Fig. 2H) were used to validate the dimeric structure of FSP1, enzyme kinetic assays (Supplementary Table 3) was used to provide more detailed information about oxidoreductase activity of FSP1, IR spectroscopy (Supplementary Fig. 9) and FENIX assay (Supplementary Fig. 13B) were used to further explain the characteristic of 6-hydroxy-FAD. Also, the results of additional hydrogen peroxide fluorometric experiment and NAD(P)H oxidation assays in aerobic or anaerobic condition (Supplementary Figure 11, Supplementary Table 3) were included to support our conclusions in the revision.

Major Comments

1) Although the dimeric structure is potentially interesting, additional characterization is required to support the conclusion that this is physiologically relevant.

- In the gel filtration chromatography data (Figure S3A), the hFSP1^{ΔCTD} runs at a larger size than hFSP1. This does not make sense. The authors argue that hFSP1 only dimerizes under conditions of high NADH and that this dimerization is CTD dependent, but the gel filtration is not consistent with this claim.

Response: Thanks! Following your suggestions, we have done the dynamic light scattering (DLS) of FSP1 proteins at two different temperature condition (8 °C or 25 °C) (Supplementary Figure 3 in the revision), and the more detailed DLS results showed that NADH did not mediate the dimerization of FSP1 proteins, suggesting that our original opinion “both cFSP1 and hFSP1 treated with excessive NADH were dimers” in the original manuscript is inaccurate. We also corrected the inaccurate interpretation, which was caused by ignoring the effects of the scanning temperature, incubation time and different purified hFSP1 proteins on the results. The sentence has been corrected to “In addition, unlike AIF^{27,28}, the dimerization of both cFSP1 and hFSP1 are not mediated by NADH” in the revision (Page 6 line 142-144 in the marked revision).

Our results demonstrated that cFSP1 existed as a dimer in solution, and the oligomeric state of hFSP1 is dynamic and diverse *in vitro*. DLS results showed that the purified hFSP1 can sometimes initially exist as a monomer at low temperature (Supplementary Figure 3C) and gel filtration chromatography analysis was performed at 4 °C, we thought that hFSP1 behaved as mostly monomer in gel filtration chromatography (Supplementary Figure 3A). DLS results showed that both cFSP1^{ΔCTD} (residues 1-318) and hFSP1^{ΔCTD} (residues 1-316) are dimer (Supplementary Figure 3C). Therefore, it is reasonable that the hFSP1^{ΔCTD} (dimer) runs at a larger size than hFSP1 (monomer) in the gel filtration chromatography at 4 °C.

Sorry for the misleading interpretation “this dimerization is CTD dependent” induced by the unclear description. In the original manuscript (Page 5 line 132-134), we have mentioned that the deletion constructs of CTD (cFSP1^{ΔCTD}, residues 1-318, or hFSP1^{ΔCTD}, residues 1-316) mainly exist as a dimer in solution. It is obvious that the dimerization of FSP1^{ΔCTD} is not dependent on the CTD and FSP1^{ΔCTD} form a non-functional dimer by a different mode. Indeed, we intended to point out that CTD is involved in FSP1 **functional** homodimerization and necessary for the **functional** dimer of FSP1 but **not for** all dimer. For clarity, we have named the dimer of FSP1^{ΔCTD} as “non-functional dimer”, and reorganized the main text to make it clear in the revision (Page 11 line 203-216 in the marked revision).

- Analysis of the Δ CTD mutant by dynamic light scattering in the presence and absence of NADH would be useful to support the dimerization conclusions.

Response: DLS results showed that the deletion constructs of CTD (cFSP1^{ΔCTD}, residues 1-318, or hFSP1^{ΔCTD}, residues 1-316) mainly exist as a dimer in the presence and absence of NADH, indicating that the non-functional dimer is formed, and may not be depended on NADH.

- hFSP1 is active at much lower NADH concentrations than are required for induced dimerization. Doesn't this argue that the dimerization is not required for its activity?

Response: Sorry for our original inaccurate interpretation “both cFSP1 and hFSP1

treated with excessive NADH were dimers”, the more detailed DLS results showed that NADH did not mediate the dimerization of FSP1 proteins (Supplementary Figure 3c), suggesting that NADH is not required for the dimerization of cFSP1 and hFSP1. Our results confirmed that cFSP1 existed as a dimer in solution, and the oligomeric state of hFSP1 is dynamic and diverse *in vitro*. The gel filtration chromatography showed that hFSP1 behaved as mostly monomer (Supplementary Figure 3A), DLS results showed that hFSP1 can sometimes start as a monomer, then automatically and fast dimerize at low temperature, and hFSP1 normally existed as a dimer at either low or room temperature (Supplementary Figure 3C). Although the slight difference between gel filtration and DLS results allows a little possibility that hFSP1 may be monomer during catalysis, our data suggested that hFSP1 functions as a dimer and the dimerization should be required for its activity, which is further supported by co-IP results showing that hFSP1 formed a dimer with itself in cells (Fig.2H).

- No data is provided to demonstrate that FSP1 dimerizes in cells. While the Δ CTD mutant does not rescue ferroptosis, this may be due to effects on dimerization or on its coordination of CoQ and its oxidoreductase activity. Does FSP1 dimerize in cells? This could be addressed by several approaches, such as FRET or CoIP.

Response: We have done the western blot analysis of co-IP experiments using HEK293T cells co-transfected with hFSP1-Flag and hFSP1-Myc. The co-IP results clearly showed that hFSP1 dimerizes in cells (Figure 2H). Although the deletion constructs of CTD (cFSP1 ^{Δ CTD}, residues 1-318, or hFSP1 ^{Δ CTD}, residues 1-316) can also be dimerized, the FSP1 ^{Δ CTD} forms a non-functional dimer and the dimerization mode of FSP1 ^{Δ CTD} is different from that of FSP1 full-length. As described above, the FSP1 ^{Δ CTD} mutant does not rescue ferroptosis because CTD plays multiple roles affecting the functional dimerization, membrane association, CoQ binding and oxidoreductase activity.

- Can the authors comment on previous studies showing that C-terminally tagged GFP FSP1 is functional and able to suppress ferroptosis? Would this large C-terminal tag

be expected to disrupt the putative dimer and if so would this suggest that the dimer is not necessary?

Response: We have carried out the expression and purification of His6-SUMO-hFSP1-eGFP (N-terminally tagged His6-SUMO and C-terminally tagged eGFP) from *E. coli*. SDS-PAGE, gel filtration chromatography (Superdex 200 Increase 10/300 GL column used in this analysis was different from that used in the main text) and DLS results showed that His6-SUMO-hFSP1-eGFP protein was unstable and oligomeric (dimer or trimer) (Point-to-Point Figure 1). Based on its oxidoreductase activity, we speculate that His6-SUMO-hFSP1-eGFP is functional and able to suppress ferroptosis in human cells, consistent with previous study (Nature, 2019, 575(7784):688-692). Although we have not confirmed that His6-SUMO-hFSP1-eGFP protein is mostly dimer because of the poor stability *in vitro*, these results support that C-terminal tagged GFP would function as oligomeric, and the dimer maybe also necessary.

Point-to-Point Figure 1. Gel filtration chromatography on a Superdex 200 Increase 10/300 GL column, dynamic light scattering and NADH consumption assay (340 nm) of hFSP1. The left insert in gel filtration chromatography shows the SDS-PAGE.

Taken together, our results showed that cFSP1 existed as a dimer, most of DLS results

displayed that hFSP1 usually existed as a dimer in solution at either 8 °C or 25 °C (Supplementary Figure 3C). Although the slight difference between gel filtration and DLS results of hFSP1 protein cannot rule out the possibility that hFSP1 may be monomer during catalysis, our data strongly suggested that hFSP1 functions as a dimer similar to cFSP1, which is further supported by co-IP results showing that hFSP1 formed a dimer with itself in cells (Fig.2H). Indeed, similar homodimer structures were observed in crystals of AIF, Ndi1 and bacterial NDH-2 enzymes. Therefore, the homodimer observed in cFSP1 structures should be physiologically relevant and the CTD-mediated dimerization should be required for FSP1 function. We have corrected the corresponding text in the revision (Page 6 line 144 - Page 7 line 157, and Page 11 line 203-216 in the marked revision).

2) To support the conclusion that hFSP1 oxidizes NADH and NADPH with similar efficiency (line261), full enzyme kinetics should be performed (i.e., Michaelis Menten kinetics). Similarly, enzyme kinetics should also be performed comparing the 6-hydroxy-FAD and FAD hFSP1 to support the conclusions (lines 340-342).

Response: Thanks for your suggestions, we have performed enzyme kinetic assays (Supplementary Table 3 in the revision). The results showed that hFSP1 oxidized NADH and NADPH with similar kinetics (K_m , K_{cat}) in the combinations with CoQ1, and K_m Value for CoQ1 reduction due to NADH oxidation was similar to that due to NADPH oxidation. Although 6-hydroxy-FAD-reconstituted hFSP1 exhibited lower affinities (K_m) for NADH and CoQ1 than FAD-reconstituted hFSP1, the NADH oxidation activity (K_{cat}) of 6-hydroxy-FAD-reconstituted hFSP1 was higher than that of FAD-reconstituted hFSP1. Therefore, the results of enzyme kinetic assays provided evidence to support our conclusions (Page 14 line 289-293 and Page 19 Line 377-381 in the marked revision).

3) It is absolutely required that peaks are assigned in MS/MS spectra. In Figure 4D, assign fragments to the MS/MS spectrum and show a comparison with pure FAD to rule out that FAD did not just form a water adduct during ionization. In Figure S8,

assign peaks to the carbon atoms in the structure and compare to native FAD.

Response: Yes, we have assigned the fragments to the MS/MS spectrum, and added the MS/MS spectrum of commercial FAD (Sigma_Aldrich), and updated the Figure 4D. We have also completed the assignment of the peaks in ¹H NMR spectrum, added the ¹H NMR spectrum of commercial FAD, and updated the results in Supplementary Figure 8. These data rule out that 6-hydroxy-FAD is a water adduct of FAD.

4) It is mentioned isomer forms of FAD (hydroxyl v ketonyl) could not unambiguously distinguished. Could this be distinguished by IR spectroscopy (rule out carbonyl species)? This is an important point to support the conclusions related to 6-hydroxy-FAD.

Response: Thanks for your suggestion. We have performed the IR spectroscopy analyses of the commercial FAD and the purified 6-hydroxy-FAD (Supplementary Figure 9). There are no obvious peaks of O–H in phenolic hydroxyl group at ~ 3600 cm⁻¹ or the characteristic peak of C=O in ketonic carbonyl group at ~1710 cm⁻¹. Although the existence of hydroxyl at the 6 position cannot be confirmed, the possibility of carbonyl at the 6 position has been at least partially ruled out. Therefore, these data validate our original speculation that the exact nature at the 6 position of the modified flavin is hydroxyl (Page 18 line 366-368 in the marked revision).

5) It is stated that “the amount of the generated H₂O₂ with CoQ was obviously higher than that without CoQ, which is small and as a cumulative result over the whole analysis period (Fig. 4G), consistent with the NADH oxidation assay result that no obvious NADH 370 consumption was observed in the absence of CoQ”. It is unclear why CoQ required to generate H₂O₂? In this scenario CoQ is present and FSP1 would employ it as the preferred substrate over oxygen. Unclear under which conditions CoQ vs oxygen would be employed. Enzyme kinetics could be useful to support the conclusions.

Response: As concerned, we have performed additional hydrogen peroxide fluorometric experiment, NAD(P)H oxidation and enzyme kinetics assays in aerobic

or anaerobic condition to make the conclusions more clear (Supplementary Figure 11, Supplementary Table 3 in the revision). Due to the limitations of the experimental conditions, the kinetic for oxygen was not measured. In the anaerobic condition, FSP1 effectively reduced CoQ1 (Supplementary Figure 11) with a higher K_{cat} and lower apparent K_m compared to those in the aerobic condition (Supplementary Table 3). These data confirmed that FSP1 generate H_2O_2 in NAD(P)H and oxygen dependent manner, and CoQ1 is actually required to generate H_2O_2 at a low concentration of FSP1. Considering the oxygen content of solution (typically >8 mg/L, 250 μ M) and no obvious NAD(P)H consumption in absence of CoQ1 under aerobic condition, and the high affinity for CoQ1, it would be reasonable that FSP1 reduce CoQ as the preferred electron acceptor over oxygen. Collectively, these data consolidate our observations that FSP1 can effectively reduce CoQ when CoQ acts as single electron acceptor substrate, and can also effectively reduce both CoQ1 and oxygen when CoQ1 and oxygen exist together, but cannot effectively reduce oxygen as only electron acceptor *in vitro*.

As for the reason why CoQ is required to effectively generate H_2O_2 , we speculate that there may be two main reasons. First, FSP1 cannot effectively transfer electrons from NAD(P)H to oxygen as only electron acceptor due to lower affinity; Second, CoQ can effectively promote the electron transfer from NAD(P)H to oxygen by acting as a CoQ-FAD intermediate complex, which is similar to the yeast NDH-2 (Ndi1) (Nature. 2012, 491(7424):478-482). Currently, these explanations cannot be clearly confirmed by our data at hand, it does not affect our main conclusions in this study. Therefore, in order to avoid overspeculation and confusion, we have added “Limitations of the study” section in the revision to mention this issue (Page 23 line 505-507 in the marked revision).

6) Based upon data using DPPH, the authors conclude that 6-hydroxy-FAD is not a free radical scavenger. DPPH is not a radical trap / reporter. This is not the correct experiment to test whether it scavenges lipid peroxy radicals. See Shah et al., Cell Chem Biol 2019 (<https://doi.org/10.1016/j.chembiol.2019.09.007>). This reference

(which is cited) makes a point that DPPH should not be used to derive / measure lipid radical scavenging activities. All DPPH assays should be reevaluated, and other methods employed.

Response: Thanks for your suggestions. DPPH assays have been deleted in the revision, and FENIX assays have been performed to make the results convinced (Supplementary Figure 13B). The new results showed that increasing the concentration of 6-hydroxy-FAD led to further suppression of the oxidation rate and a corresponding increase in the inhibited period, suggesting that 6-hydroxy-FAD is a potent radical-trapping antioxidant (RTA) in lipid membranes, slightly weaker than another water-soluble RTA BH4. We have corrected the information in the main text in the revision (Page 21 line 441-447 in the marked revision)..

Minor Comments

1) A brief statement of why chicken FSP1 was analyzed vs human FSP1 would be helpful.

Response: In the study, we initially attempted to obtain the crystal of hFSP1, but were not successful. Therefore, cFSP1 was purified and crystallized. The statement has been added in the revision (Page 3 line 71-73 in the marked revision).

2) Figure 2D, the overlaid curves are not easily distinguished. Please use different colors / shapes.

Response: Thanks. Done.

3) Figure 2G, western blot is of unacceptable quality.

Response: We have re-performed the Western blot and re-uploaded this data in the revision.

4) Figure 4E, use an offset of the chromatograms and consider quantifying the peaks. As shown, it is very difficult to compare relative peak intensities.

Response: Thanks. Done.

Reviewer #2

FSP1 is a recently discovered glutathione-independent ferroptosis suppressor, but its underlying structural mechanism has remained unknown. In this manuscript, the authors report the crystal structure of chicken (*Gallus gallus*) FSP1 (cFSP1) in the substrate-free and ubiquinone-bound states. The authors found that FSP1 has a FAD-binding domain, a NAD(P)H-binding domain, and a unique C-terminal domain. The C-terminal domain mediates functional dimerization of FSP1 and participates in the active site formation. FSP1 catalyzes the formation of 6-hydroxy-FAD, and the 6-hydroxy-FAD containing cFSP1 is catalytically active. Importantly, in the absence of FSP1, 6-hydroxy-FAD, but not FAD, can rescue ferroptosis. Therefore, this study establishes 6-hydroxy-FAD as an active cofactor of FSP1 and a potent ferroptosis inhibitor. Overall, these are significant findings that would merit the publication in Nature Communications, providing the following issues are adequately addressed.

Major:

How 6-hydroxy-FAD inhibits ferroptosis needs be better explained. Currently, this molecule is not connected with any known ferroptosis resistance mechanism. How does this molecule inhibit ferroptosis in the absence of FSP1?

Response: Thanks for your positive comments. As mentioned in response to Reviewer 1's comments, we have performed the FENIX assays (Cell chemical biology, 2019, 26(11), 1594-1607 e1597) to determine lipid radical-trapping activities of 6-hydroxy-FAD (Supplementary Figure 13B in the revision). The results showed that increasing the concentration of 6-hydroxy-FAD led to further suppression of lipid peroxidation rate and a corresponding increase in the inhibited period, suggesting that 6-hydroxy-FAD is a potent radical-trapping antioxidant (RTA) in lipid membranes, slightly weaker than another water-soluble RTA BH4. Indeed, the cellular uptake experiments showed that 6-hydroxy-FAD alone can rescue the resistance of HT1080 hFSP1KO cells to RSL3 (Fig. 4H) and promote the resistance of HT1080 wild type

cells to RSL3 (Supplementary Fig. 13A). The NAD(P)H oxidation activity of 6-hydroxy-FAD-containing hFSP1 is higher than that of FAD-containing hFSP1. Collectively, these data indicated that 6-hydroxy-FAD actually acts a dual role as an active cofactor for FSP1 and a potent antioxidant *in vitro* and **may inhibit ferroptosis in the absence of FSP1 by acting as a RTA**. However, we could not detect 6-hydroxy-FAD in the lysate from HT1080 hFSP1^{KO} or HEK293T cells overexpressing hFSP1 (Supplementary Fig. 14) or HT1080 wild type or HT1080 hFSP1^{KO} cells treated with 6-hydroxy-FAD (Point-to-Point Figure 2) by LC-MS. Although we speculate that 6-hydroxy-FAD may function as a cofactor of FSP1 and antioxidant involved in the ferroptosis inhibition in cells at levels below our detection limit, we cannot rule out the possibility that 6-hydroxy-FAD may not be stably stored (e.g. immediately converted into other compound) in human cells and the feeding 6-hydroxy-FAD blocks ferroptosis probably by other unclear mechanism. This should be pursued in the near future. We have reorganized the main text in the revision (Page 21 line 439-456 in the marked revision) to make it clear and avoid overspeculation. In addition, we have added “Limitations of the study” section in the revision to mention this issue (Page 23 line 507-510 in the marked revision).

Point-to-Point Figure 2. HPLC-MS analysis (UV 254 nm) of cell lysate. HPLC chromatograms (A), MS extracted ion chromatogram (B, m/z 786.1644; C, m/z 802.1593) of the boiled hFSP1 protein (black line), HT1080 wild type cells treated with 10 μ M 6-hydroxy-FAD for 1h (red line) or 2h (green line), FSP1^{KO} HT1080 cells treated with 10 μ M 6-hydroxy-FAD for 1h (blue line) or 2h (yellow line). FAD (calculated mass [M+H]⁺, 786.1644) and 6-hydroxy-FAD (calculated mass [M+H]⁺, 802.1593) were eluted at 10.5 min and 11.7 min, respectively.

Can the authors capture 6-hydroxy-FAD in their crystal structure? Since the authors

can isolate 6-hydroxy-FAD, this is worth a shot, as this would provide strong support for the proposed mechanism.

Response: It is a good suggestion to try to capture 6-hydroxy-FAD in the crystal structures, which would obviously be helpful for our conclusions. However, so far, we have not obtained the crystal structure of FSP1 in complex with 6-hydroxy-FAD by using various trials as follows. 1. Crystallization of 6-hydroxy-FAD-reconstituted FSP1 proteins (hFSP1, hFSP1^{ΔN}, cFSP1 or cFSP1^{ΔN}): we failed to obtain qualified crystals (crystals of 6-hydroxy-FAD-reconstituted cFSP1^{ΔN} were too small and very difficult to be optimized). 2. Crystallization of 6-hydroxy-FAD-reconstituted and methylated cFSP1^{ΔN}: the methylation of lysine residues leads to a large amount of precipitated protein, we failed to obtain the crystal. 3. Crystallization of cofactor-removed hFSP1^{ΔN} or cFSP1^{ΔN}: we failed to obtain the crystal. 4. Soaking the crystal of methylated cFSP1^{ΔN} in the cryoprotectant solution supplemented with 6-hydroxy-FAD: we have collected several datasets and solved the structures, but the observed cofactor in the crystal structures was still FAD, not 6-hydroxy-FAD. It is difficult for us to capture 6-hydroxy-FAD in the crystal structures in a short time. Anyway, our data can support the existing conclusions.

Minor:

(line 121): “though it is partially disordered and displayed weak density” should be “though it is partially disordered and displays weak density”.

Response: Thanks. Done.

Reviewer #3

In the present work Lv et al., report on the crystal structure of FSP1 (previously known as AIFM2) unbound and bound to one of its substrates, ubiquinone. The study appears to be carefully conducted and the report of the FSP1 structure is timely and of interest as it could catalyse a deeper understanding of the biology of FSP1 as well as helping to understand the mode of actions of FSP1 inhibitors.

The current revision does deep into the specifics of the structural work as I don't feel capable to judge these aspects. Therefore I have only limited my comments to particular aspect of the biology of FSP1 and ferroptosis.

One of aspect that I find not entirely convincing is regarding the role of 6-hydroxyFAD (6OH-FAD) in ferroptosis. Based on the observation that 6OH-FAD is formed in a H₂O₂ dependent manner during catalysis in vitro, the authors posit that this intermediate could be formed in cells and could is involved in protecting cells from ferroptosis. While in Figure 4H they show that 6OH-FAD can suppress RSL3 induced ferroptosis this might not be entirely surprising given the presence of the 6-OH group that could work as a direct antioxidant (the DPPH assay provided in SI is not sufficient to exclude this). Still, I find remarkable that 6OH-FAD can be directly taken up by cells; to the best of my knowledge FAD is, if at all, very poorly taken up by cells. The question remains, if cells can produce 6OH-FAD at sufficient high levels to make it a relevant inhibitor. Did the authors attempt to measure it in cells and what's the contribution of FSP1 to its levels? I don't think this disqualifies the work but some discussion and potentially toning down some of the conclusions seems justifiable.

Response: Thanks for your valuable suggestions.. We have deleted the text about DPPH assays that are not a reliable radical trap / reporter, and performed the FENIX assays to determine lipid radical-trapping activities of 6-hydroxy-FAD (Supplementary Figure 13B in the revision). The results suggested that 6-hydroxy-FAD is a potent radical-trapping antioxidant (RTA) in lipid membranes probably due to its 6-OH group (as reviewer's mentioned), slightly weaker than another water-soluble RTA BH4.

As the role of 6-hydroxy-FAD in ferroptosis inhibition in human cells, we agree to your comments. The LC-MS experiments were performed to measure the levels of 6-hydroxy-FAD in human cells and to detect if 6-hydroxy-FAD can be directly taken up by cells. Interestingly, 6-hydroxy-FAD has not been detected in the lysate from HT1080 hFSP1^{KO} or HEK293T cells overexpressing hFSP1 (Supplementary Fig. 14) or HT1080 wild type or HT1080 hFSP1^{KO} cells treated with 6-hydroxy-FAD (Point-to-Point Figure 2) by LC-MS. However, feeding 6-hydroxy-FAD can actually

restore ferroptosis resistance in HT1080 hFSP1^{KO} cells (Fig. 4H) and promote the ferroptosis resistance of HT1080 wild type cells (Supplementary Fig. 13A), suggesting that 6-hydroxy-FAD should be taken up by human cells to elicit its anti-ferroptosis function. Therefore, although 6-hydroxy-FAD acts a dual role as an active cofactor for FSP1 and a potent antioxidant *in vitro* and may play the same role in the ferroptosis inhibition in human cells at levels below our detection limit, we cannot rule out the possibility that 6-hydroxy-FAD may not be stably stored (e.g. immediately converted into other compound) in human cells and feeding 6-hydroxy-FAD blocks ferroptosis probably by other unclear mechanism. This should be pursued in the near future. So, following your suggestions, we have reorganized the information in the main text in the revision (Page 21 lines 439-456 in the marked revision) to make it clear and avoid overspeculation. In addition, we have added “Limitations of the study” section in the revision to mention this issue (Page 23 line 507-510 in the marked revision).

REVIEWERS' COMMENTS

Reviewer #1 (Remarks to the Author):

The authors have carefully address my concerns and I recommend publication. Congratulations on this exciting and important manuscript.

Reviewer #2 (Remarks to the Author):

In this revision manuscript, the authors have made substantial efforts to address the reviewer critiques, though the failure to detect 6-hydroxyl-FAD in cells at a sufficient concentration is a major concern that undermines the entire study. As a result, this work is premature for publication at Nature Communications.

Reviewer #3 (Remarks to the Author):

The authors have satisfactorily addressed my comments - congratulations on this interesting and important work.

Responses in Point-by-Point to the Comments of Referees

Reviewer #1 (Remarks to the Author):

The authors have carefully address my concerns and I recommend publication. Congratulations on this exciting and important manuscript.

Response: Thanks.

Reviewer #2 (Remarks to the Author):

In this revision manuscript, the authors have made substantial efforts to address the reviewer critiques, though the failure to detect 6-hydroxyl-FAD in cells at a sufficient concentration is a major concern that undermines the entire study. As a result, this work is premature for publication at Nature Communications.

Response: Thanks. Our results have confirmed that FSP1 can generate 6-hydroxy-FAD *in vitro*, which acts a dual role as an active cofactor for FSP1 and a potent antioxidant *in vitro* (Fig. 5, Supplementary Fig. 9, 10, 11 and 14). Feeding 6-hydroxy-FAD can actually restore ferroptosis resistance in HT1080 hFSP1KO cells (Fig. 5e) and promote the ferroptosis resistance of HT1080 wild type cells (Supplementary Fig. 14a). However, as you mentioned, we have not detected 6-hydroxy-FAD in human cells. The exact role of 6-hydroxyl-FAD in human cells needs future study to elucidate. We have mentioned this issue in the “Limitations of the study” section of the paper.

Reviewer #3 (Remarks to the Author):

The authors have satisfactorily addressed my comments -congratulations on this

interesting and important work.

Response: Thanks.